# Milk Exosomal microRNAs: Postnatal Promoters of β Cell Proliferation but Potential Inducers of β Cell De-Differentiation in Adult Life

**DOI:** 10.3390/ijms231911503

**Published:** 2022-09-29

**Authors:** Bodo C. Melnik, Gerd Schmitz

**Affiliations:** 1Department of Dermatology, Environmental Medicine and Health Theory, University of Osnabrück, D-49076 Osnabrück, Germany; 2Institute for Clinical Chemistry and Laboratory Medicine, University Hospital of Regensburg, University of Regensburg, D-93053 Regensburg, Germany

**Keywords:** beta-cell, diabetes mellitus, milk exosome, microRNA, beta-cell de-differentiation, beta-cell identity, proliferation

## Abstract

Pancreatic β cell expansion and functional maturation during the birth-to-weaning period is driven by epigenetic programs primarily triggered by growth factors, hormones, and nutrients provided by human milk. As shown recently, exosomes derived from various origins interact with β cells. This review elucidates the potential role of milk-derived exosomes (MEX) and their microRNAs (miRs) on pancreatic β cell programming during the postnatal period of lactation as well as during continuous cow milk exposure of adult humans to bovine MEX. Mechanistic evidence suggests that MEX miRs stimulate mTORC1/c-MYC-dependent postnatal β cell proliferation and glycolysis, but attenuate β cell differentiation, mitochondrial function, and insulin synthesis and secretion. MEX miR content is negatively affected by maternal obesity, gestational diabetes, psychological stress, caesarean delivery, and is completely absent in infant formula. Weaning-related disappearance of MEX miRs may be the critical event switching β cells from proliferation to TGF-β/AMPK-mediated cell differentiation, whereas continued exposure of adult humans to bovine MEX miRs via intake of pasteurized cow milk may reverse β cell differentiation, promoting β cell de-differentiation. Whereas MEX miR signaling supports postnatal β cell proliferation (diabetes prevention), persistent bovine MEX exposure after the lactation period may de-differentiate β cells back to the postnatal phenotype (diabetes induction).

## 1. Introduction

Recent studies have associated exosomal microRNAs (miRs) with the pathogenesis of type 2 diabetes mellitus (T2DM) [1]. Human milk and cow milk are rich sources of milk exosomes (MEX), which deliver their miR cargo into the systemic circulation and various tissues of the milk recipient. Under physiological conditions, human MEX may also reach the pancreatic β cells, promoting β cell growth and mass expansion during the period of lactation. Physiologically, MEX miR exposure and signaling terminates after weaning. In contrast to all other mammals, humans of Western societies are throughout their entire life exposed to MEX delivered by the consumption of pasteurized cow milk. It is the intention of this review (1) to elucidate the potential MEX miR-mediated effects of human milk on β cell homeostasis, proliferation, mass expansion, and suppression of glucose-stimulated insulin secretion (GSIS) during the physiological period of breastfeeding and (2) to investigate the potential impact of persistent bovine MEX miR exposure on adult β cell homeostasis. Translational evidence indicates that persistent bovine MEX miR signaling by continued consumption of pasteurized cow milk may be an important and overlooked mechanism promoting β cell de-differentiation back to the postnatal phenotype with increased mTORC1/c-MYC-driven β cell proliferation, leading to increased endoplasmic reticulum (ER) stress-induced β cell apoptosis and compromised GSIS.

To understand the impact of MEX miR signaling in the pathogenesis of T2DM, we must learn more about the physiological interaction of human MEX with postnatal β cells of the infant. The postnatal period is a critical window for epigenetic programming [2,3,4]. During the postnatal period, the endocrine pancreas expands to reach an appropriate adult β cell mass, and the insulin-secreting cells finalize their functional maturation [5,6,7]. Multiple signaling pathways orchestrate the proliferation and functional maturation of β cells and are vital to ensuring adequate insulin requirements in adulthood with the transition from infant to adult nutrition [8]. Early postnatal nutrition is currently the focus of diabetes research, as inappropriate nutrition early in life poses a risk for developing diabetes in adulthood, which can affect the β cells of the offspring even over two generations [7]. Recent evidence indicates that miRs play a crucial role for β cell proliferation, mass expansion, maturation, and failure [1,9,10,11]. Of note, postnatal β cell maturation is associated with islet-specific miR changes induced by nutrient shifts at weaning [12]. Milk transfers a vast spectrum of extracellular vesicles (EVs), including bioactive MEX and their miRs to the infant participating in metabolic and epigenetic regulations [13,14,15,16,17,18,19,20]. MEX miRs are highly conserved between mammals [15,21], survive the harsh conditions of the gastrointestinal tract [22,23,24], are taken up by endocytosis [25,26], are bioavailable [27,28], reach the systemic circulation [29], and enter the cells and peripheral tissues [30,31,32,33,34,35].

The translational evidence provided in this review supports the view that MEX miRs may function as potential maternal signalosomes inducing β cell proliferation and mass expansion, whereas weaning, the period of life associated with the termination of physiological MEX miR exposure, may represent the key signal for switching from β cell proliferation to β cell maturation with GSIS [36]. Continuous oral exposure of humans to bovine MEX miRs after the weaning period is an unnatural behavior that may promote β cell de-differentiation, loss of β cell identity, and eventually early β cell apoptosis, thus promoting and accelerating the onset of T2DM in societies consuming cow milk.

## 2. Exosome Crosstalk with Pancreatic β Cells

Pancreatic β cells maintain a crosstalk with different exosomes either derived from neighboring islet β cells [37,38], islet macrophages [39,40], T-lymphocytes [41], or exosomes derived from peripheral cells including adipocytes and adipose-tissue macrophages [42], muscle cells [43], placenta cells [44,45], and mesenchymal stem cells [46]. This supports the concept of exosomal miR-mediated communication between and towards β-cells [47] (Figure 1).

For instance, human T lymphocyte-derived exosomes (T-EX), which transfer miR-155, induce apoptosis of β cells and promote type 1 diabetes mellitus (T1DM) in mice [41]. Exosomes from insulin-resistant muscles influence gene expression and proliferation in murine β cells [43]. Exosomes from obese adipose tissue are harmful for human β cells and exhibit a four-fold increased expression of miR-155 [42]. Placenta-derived exosomes are involved in the pathogenesis of gestational diabetes mellitus (GDM) [44,45]. Recently, Gao et al. [46] showed that adipose tissue macrophage-derived exosomes (ATM-EX) from obese mice are highly enriched in miR-155, suppress insulin secretion, and enhance β cell proliferation. In contrast, knockout of miR-155 attenuated the adverse effects on the β cell of ATM-EX of obese adipose tissue. Apparently, the miR-155/MAFB axis plays a critical role in controlling β cell homeostasis.

In contrast, He et al. [47] investigated the impact of bone marrow mesenchymal stem cell-derived exosomes (MSC-EX) administrated to T2DM rats and high-glucose-treated primary islets. Their results showed that MSC-EX and especially miR-146a therein reversed diabetic β cell de-differentiation and improved β cell insulin secretion [47]. Furthermore, human umbilical MSC-EX inhibited streptozotocin (STZ)-induced β cell apoptosis and restored insulin-secretion in a model of T2DM [48].

Non-coding RNAs from tissue-derived small extracellular vesicles (EVs) play an important role in diabetes research [1,49]. The beneficial as well as adverse effects of exosome traffic to β cells and their impact on β cell homeostasis have been the recent focus of investigations [1,50,51,52,53,54,55]. As MEX and their miR cargo reach the systemic circulation and tissues [29,30,31,32,33,34,35], they may as well communicate with the neonatal β cell during the period of lactation and later on may affect the adult islet organ via consumption of pasteurized cow milk [36] (Figure 1).

Thus, five specific questions arise: (1) What is the physiological impact of human MEX on neonatal β cells during the breastfeeding period? (2) Does a MEX-deficient infant formula feeding impair adequate β cell maturation, increasing the risk of T2DM later in life? (3) Do modifications of maternal MEX composition or iatrogenic interventions during the perinatal period affect the MEX miR composition and β cell maturation of the infant? (4) Does the persistent consumption of pasteurized cow milk (bovine MEX exposure) over a lifetime de-differentiate β cells back to the postnatal proliferative/immature phenotype? (5) Do bovine MEX eventually synergize with the proinflammatory effects of ATM-EX derived from obese adipose tissue? This review intends to provide answers for these issues, based on evidence from published literature.

## 3. Postnatal MEX miR Signaling Promotes Pancreatic β Cell Proliferation

The functional maturation of insulin-secreting β cells is initiated before birth and is completed during postnatal life. Appropriate postnatal β cell proliferation has a pivotal impact on the acquisition of an adequate functional β cell mass and on the capacity of the islets to meet and adapt to insulin needs later in adult life. It has been found that pancreatic β cell expansion and functional maturation during the birth-to-weaning period plays an essential role in the adaptation of plasma insulin levels to metabolic needs. These events are driven by epigenetic programs, triggered by growth factors, hormones, and nutrients [11]. A major source of these humoral factors is maternal milk, which also transfers MEX and their miRs to the infant [11,12,13,14,15,16,17,18,19,20]. MEX miRs apparently play a crucial role for epigenetic programming of β cell proliferation during the breastfeeding period. During postnatal development, β cells are highly proliferative and their expansion contributes to a substantial increase in β cell mass [56,57,58,59,60,61].

### 3.1. The Role of MEX miRs in AKT-mTORC1-Mediated β Cell Proliferation

A central signaling pathway for β cell proliferation is the AKT-mTORC1 pathway [62,63,64,65,66,67,68]. mTORC1 regulates β cell proliferation, cell size, and mass [67]. Furthermore, mTORC1 plays an essential role in β cell adaptation to significant losses of β cells, as shown in 60% partial-pancreatectomy mice and for early compensatory β cell proliferation via the cyclin D2 pathway [69].

Milk and its signaling components have been identified as a unique signal transduction system activating mTORC1 [70], which is the central kinase of metabolism orchestrating growth signals and nutrient availability for cell growth and anabolism [71,72,73,74,75,76]. Importantly, target gene prediction of human, bovine, and porcine MEX miRs mainly concentrate on PI3K-AKT-mTORC1 pathway activation [28,77,78,79].

Somatolactogenic hormones including growth hormone (GH), prolactin (PRL), and insulin-like growth factor-1 (IGF-1) and their receptors (GHR, PRLR, and IGF1R) are critically involved in β cell growth and survival [80,81]. β cells endogenously express GHR, Janus kinase 2 (JAK2), and IGF1R, respectively [82]. Ma et al. [82] detected a GHR-JAK2-IGF1R protein complex in β cells upon GH stimulation in rodent β cells. Their findings strongly suggest a signaling relationship between GH/GHR and IGF-1/IGF1R systems in β cells. IGF1R may serve as a proximal component of GH/GHR signaling, contributing to enhancement of AKT activation, promoting β cell proliferation [82]. In accordance, it has been demonstrated in breast cancer cells that PRL and IGF-1 synergistically activate AKT, leading to increased cell proliferation [83]. Notably, prolactin augments tyrosine phosphorylation of IGF1R on cotreatment with IGF-1 [83]. PRL cotreatment with IGF-1 augmented a two-fold higher IGF1R phosphorylation than IGF-1 alone [84]. Furthermore, PRL diminished IGF-1-induced IGF1R internalization. GH and PRL via activation of JAK2 stimulated tyrosine phosphorylation of insulin receptor substrate (IRS)-1, IRS-2, and IRS-3 [85]. As shown in neonatal rats, PRL stimulates the phosphorylation of IRS-1 and IRS-2, activating PI3K [86], which are crucial upstream events that activate AKT. Of importance, PRL signaling increases pancreatic islet mass during pregnancy [87]. Activation of the PI3K/AKT pathway was not only essential for IGF-1/glucose-induced β cell proliferation [88] but also important for promoting β cell survival and maintaining β cell mass [89,90]. Thereby, IRS-2 expression appeared to be the most critical [91]. Compared to IRS-1, IRS-2 has been demonstrated to play a major role in IGF1R-mediated β cell development and compensation for peripheral insulin resistance [91]. Insulin also stimulates primary β cell proliferation [92]. Insulin-stimulated mouse insulinoma cell proliferation is dependent on both PI3K/AKT and RAF1/MAPK kinase pathways [92]. Thus, PRL, GH, IGF-1, and insulin have synergistic effects potentiating IRS-2-mediated activation of PI3K, AKT, and mTORC1 [70].

MEX miR-148a attenuates the expression of phosphatase and tensin homolog (PTEN), tuberin (TSC2), and AMP-activated protein kinase (AMPK), enhancing the activation of AKT and mTORC1, which thereby enhances the expression of the transcription factors E2F1 and c-MYC, respectively (Figure 2).

### 3.2. The Role of Long Non-Coding RNA H19 for β Cell Proliferation

Remarkably, global profiling of transcripts in pancreatic islets of newborn and adult rats revealed that the transcription factor E2F1 controls the expression of the long non-coding RNA H19 (H19) in an AKT-dependent manner [93]. The expression of H19 in islets of adult rats was 303 times lower than in those of newborn rats [93]. Activated AKT induces β cell proliferation by activation of cyclin-dependent kinase 4 (CDK4)/cyclin D complex [94], which phosphorylates riboblastoma tumor suppressor protein (RB) [95], thereby activating E2F1 [93] (Figure 2). Finally, E2F1 promotes the expression of H19 [93,96]. E2F1 associates with a G-quadruplex structure at the 5′ end of the H19 coding region, enhancing H19 transcription [96]. Notably, impaired pancreatic growth, reduced β cell mass, and β cell function has been reported in E2F1^–/–^ mice [97]. In contrast, ectopic expression of E2F1 in adult β cells increased proliferation [98]. Whereas 75% of insulin-producing cells were also positive for PDX-1 in E2F1^+/+^ mice, only 30% of β cells were positive for PDX-1 in E2F1^–/–^ mice [97]. These findings in E2F1^–/–^ mice on PDX-1 expression are reminiscent of those observed in IRS-2^–/–^ mice [99], suggesting that E2F1 could directly regulate the expression of PDX-1 [97] (Figure 2).

Importantly, the transcription factor c-MYC, which is upregulated during postnatal β cell proliferation [100,101,102], directly induces H19 expression [103,104,105,106] (Figure 2).

Translational evidence indicates that H19 expression is also controlled by epigenetic mechanisms involving DNA methyltransferase 1 (DNMT1) [107] and methyl-CpG-binding protein 2 (MECP2) [108]. MECP2 attracts DNMT1 to promoter methylation sites [109]. For instance, knockdown of DNMT1 elevated H19 expression in activated hepatic stellate cells (HSCs), whereas overexpression of DNMT1 inhibited H19 expression [107]. In accordance, knockdown of MECP2 elevated H19 expression in activated HSCs, whereas MECP2 overexpression inhibited H19 expression [107]. It has recently been demonstrated that metformin, the most common antidiabetic drug, reduces H19 expression via H19 promoter methylation [110,111,112,113,114]. Thus, H19 expression is upregulated via interacting transcriptional and epigenetic mechanisms, which may involve MEX miR-mediated regulation (Figure 2).

### 3.3. H19 Acts as a Sponge of let-7 miRs, miR-29a, and miR-29b

Postnatal β-cell maturation is associated with islet-specific miR changes induced by nutrient shifts at weaning [12]. Importantly, the imprinted H19 antagonizes let-7 miRs [93,115]. H19 harbors both the canonical and non-canonical binding sites for the let-7 family of miRs, which play important roles in human development [115]. Thus, H19 modulates let-7 availability by acting as a molecular sponge. Sanchez-Parra et al. [93] demonstrated that H19 silencing decreased β cell expansion in newborns, whereas its re-expression promoted proliferation of β cells in adults via a mechanism involving let-7 and the activation of AKT. Notably, the offspring of rats fed a low-protein diet during gestation and lactation display a small β cell mass, resulting in an increased risk of developing diabetes during adulthood. Indeed, islets of newborn rats born to dams fed a low-protein diet express lower levels of H19 than those born to dams that did not receive a low-protein diet [93]. Moreover, H19 expression increases in islets of obese mice under conditions of increased insulin demand. These data suggest that H19 plays an important role in postnatal β cell mass expansion in rats and contributes to the mechanisms compensating for insulin resistance in obesity. This observation is in accordance with predicted target gene analyses showing that *IGF1R*, *INSR*, *GHR*, *PRLR*, *IRS2*, and *AKT2* are all target genes of let-7 (TargetScanHuman 8.0) (Figure 3).

Zhu et al. [116] confirmed that the 3′ UTRs of *INSR*, *IGF1R*, *IRS2*, *PIK3IP1*, *AKT2*, *TSC1,* and *RICTOR* mRNAs were targeted by let-7 for suppression and provided evidence that insulin-PI3K-mTORC1 signaling is suppressed by let-7. In contrast to let-7 suppression in postnatal proliferating β cells, overexpression of let-7 results in insulin resistance and impaired glucose tolerance [116,117]. In particular, *INSR* and *IRS2* mRNAs are direct targets of let-7 [116].

The most substantially upregulated miR during postnatal β cell maturation is miR-29b-3p [12]. Notably, in the maturing islet cells of 10-day-old rats, miR-29b-3p and miR-17-5p directly regulate the circadian gene expression [118]. Importantly, H19 acts as a sponge of miR-29a-3p [119,120] and miR-29b-3p [121,122,123,124]. By quenching endogenous miR-29a-3p, H19 induces E2F1 expression [119].

### 3.4. LIN28 and let-7

H19, which is overexpressed in proliferating β cells, acts as molecular sponge of let-7 [93,115]. In addition, the RNA binding proteins LIN28A and LIN28B selectively block let-7 biogenesis [116]. In accordance with H19, LIN28A overexpression in Min6 cells synergizes with let-7-regulated effects on β cell proliferation and survival [125]. Ablation of LIN28B during human embryonic stem cell (hESC) differentiation toward β cells led to a more mature GSIS profile and the suppression of juvenile-specific genes [126].

It has recently been shown that overexpression of let7b-5p in β cells of βlet7b-5p-transgenic mice inhibited β cell proliferation and decreased the expression of cyclin D1 and cyclin D2 [127]. In contrast, milk-mediated suppression of let-7 via upregulation of H19 may synergize with LIN28-mediated β cell proliferation during the breastfeeding period.

### 3.5. MEX miR-Mediated Regulation of p53 and miR-29b

Evidence from the literature supports an interaction between MEX miRs with the expression of p53 and miR-29b. The major miR of milk, milk fat, EVs, and MEX of all mammals is miR-148a [15,21,128,129,130]. MiR-148a-3p is the most highly expressed among all detected human MEX miRs, comprising 35.45% of total miRs [129]. The abundance of miR-148a indicates that this MEX-transported miR plays a crucial role for milk signaling during the postnatal period. MEX miR-148a directly targets *TP53* mRNA and thereby suppresses p53 [131,132], the guardian of the genome [133,134], which interacts with approximately 1/10th of human gene promoters [135]. p53 modifies the expression of target genes activating cell cycle progression (*CDKN1A*) [136,137], IGF-1 and mTORC1 signaling (*IGF1R*, *PTEN*, *TSC2*) [137,138,139,140,141], LKB1-AMPK signaling (*STK11*, *SESN1*, *SESN2*) [142,143], autophagy (*ATG5*, *BECN1*) [144,145], and apoptosis (*FOXO1A*, *FOXO3A*, *TNFRSF10B*) [137,146], promoting cell survival (*BIRC5*) [137,147,148,149]. Other highly conserved suppressors of p53 are miR-125b [150], detected in human, bovine, porcine, and sheep MEX [22,129,151,152,153,154] as well as miR-30d and miR-25 [155], identified in human and porcine MEX as well [151,154,155,156,157,158].

Furthermore, MEX miRs activate AKT [15,159] via inhibition of PTEN (miR-148a, miR-21, and miR-155) [160], enhancing p53 proteasomal degradation via AKT-mediated phosphorylation of mouse double minute 2 (MDM2) [161,162]. Mecocci et al. [163] identified MDM4 as a central node of transcriptomic regulation of cow, donkey, and goat MEX RNAs. MDM4 restricts p53 transcriptional activity and facilitates MDM2′s E3 ligase activity towards p53 [164]. In addition, increased expression of c-MYC significantly attenuated apoptosis and impaired the transcriptional activity of p53 on p21 [165]. The p53 target gene survivin (*BIRC5*), which is suppressed by p53 [149], is critically involved in the regulation of β cell mass after birth [166,167]. Targeted deletion of survivin in the pancreas results in a significant decline in β cell mass throughout the perinatal period, leading to glucose intolerance in the adult. Survivin-deficient islets show decreased cell proliferation as a result of a delay in cell cycle progression with perturbations in cell cycle proteins [167]. Thus, MEX miRs may suppress β cell p53 expression during breastfeeding-associated MEX miR signaling, whereas the disappearance of MEX miRs after weaning may enhance β cell p53 expression.

Jacovetti et al. [12] emphasized that postnatal β cell maturation is associated with islet-specific miR changes induced by nutrient shifts at weaning. In particular, a steady increase of miR-29b has been observed [12]. Importantly, miR-29b—as all other members of the miR-29 family—targets the mRNA of *SLC16A1*, which encodes the monocarboxylate transporter 1 (MCT1) [168]. Suppression of MCT1, as well as lactate dehydrogenase A (*LDHA*), ensures that glucose-derived pyruvate is efficiently metabolized by mitochondria, while exogenous lactate or pyruvate is unable to stimulate metabolism and hence results in inappropriate insulin secretion. Remarkably, transcriptional activation of miR-29 occurs in a p53-dependent manner [169]. Park et al. [170] identified miR-29 family members as positive regulators of p53. In fact, miR-29 family members activate p53 by direct suppression of p85α (the regulatory subunit of PI3K) and CDC42 (a Rho family GTPase), both of which negatively regulate p53 [170]. Thus, fading of MEX miR-148a and miR-125b after weaning may enhance p53 activity and p53-mediated upregulation of miR-29b in maturing β cells, thereby attenuating MCT1 expression. It is important to mention that miR-29b expression is also regulated by c-MYC, which suppresses miR-29b expression [171] (Figure 4). Upregulation of miR-29b is of key importance for cellular differentiation, including β cell maturation [12,172]. Enhanced expression of H19 during β cell proliferation may sponge miR-29b [120,121,122,123], which acts as positive regulator of p53 [170], thereby reducing p53 expression during breastfeeding. The disappearance of MEX miR signaling after weaning associated with a decrease of c-MYC/H19 expression may enhance β cell miR-29b expression associated with miR-29b-induced p53 expression, promoting functional β cell maturation.

Taken together, milk and especially MEX provide a gene-regulatory network of p53-targeting miRs, enhancing proliferation and cell survival [131,132,173]. We suppose that MEX reach the pancreatic β cell, and that MEX and their miRs may promote β cell proliferation during the signaling period of lactation.

### 3.6. MYC, MAX, and H19 Expression

AKT-mediated signaling via CDK4/RBp activation enhances the activity of E2F1, a critical transcription factor for the expression of c-MYC (Figure 2). AKT-mTORC1 activation via S6K1-mediated upregulation of EIF4B may also enhance the expression of c-MYC [174]. mTORC1 mediates β cell growth and expansion, whereas mTORC1-hyperactivation has been observed in pancreatic islets from animal models of T2DM, resulting in β cell loss [175]. In patients with T2DM, mTORC1 is markedly increased, while mTORC2 activity is diminished [175]. Of note, plasma levels of miR-148a of T2DM patients were significantly increased, whereas let-7d was significantly reduced [176], pointing to increased mTORC1/c-MYC/H19 signaling (Figure 2). Notably, the c-MYC/MAX/MAD transcription factor network plays a key role in growth control, cell cycle progression, differentiation, and cell death [177,178]. Important for the understanding of c-MYC function is the protein MAX, a bHLHLZ protein, which specifically dimerizes with c-MYC [177]. MAX has a dual potential as an activator and suppressor of c-MYC function [177]. In the absence of c-MYC, MAX binds to the H19 promoter proximal E-box [103], a feature previously described as a hallmark of c-MYC regulated genes [103]. According to Mao et al. [179], MAX is essential for c-MYC-dependent repression. Intriguingly, *MAX* mRNA is a direct target of miR-148a [180].

Recent evidence indicates that c-MYC and E2F1 can exert a positive combinatorial control over H19 transcription [103]. Complex interactions between DNA methylation and histone modifications regulate H19 expression, whereas the role of c-MYC is to recruit histone acyl transferase activity to unmethylated E-boxes and initiate allele-specific H19 transcription [103]. c-MYC induction of both alleles of H19 was evident in the presence of 5-azacytidine, showing that in the absence of DNA methylation, c-MYC can access and activate both alleles of H19 [103]. This is in accordance with the observation that the c-MYC/MAX complex is unable to bind to methylated E-boxes in vitro [181].

Golan-Gerstl et al. [15,160] provided substantial evidence that MEX miR-148a suppressed the expression of *DNMT1* in MEX-exposed intestinal epithelial cells. MiR-148a and miR-22, which are highly expressed in MEX of preterm infants [182], target *MECP2* [183], which guides DNMT1 to its sites of CpG methylation [109]. MEX-mediated suppression of DNA methylation-dependent silencing of developmental genes, which are critically involved in postnatal programming [184,185], may thus induce the H19 expression of β cells (Figure 2). In this regard, the antidiabetic drug metformin via methylating the H19 promoter region [111,112,113,114,115] operates in the opposite direction to MEX-mediated signaling. Metformin-induced H19 methylation may thus reduce the access of c-MYC to the H19 site of transcription regulation, which may also affect the access of E2F1. This mode of action may be central for explaining the antidiabetic and anticancer effects of metformin [186,187], which, in contrast to milk [159] is also an activator of AMPK [188] and inhibitor of mTORC1 [189].

### 3.7. FTO α-Ketoglutarate-Dependent Dioxygenase

The N6-methyladenosine (m6A) RNA modification is essential during the embryonic development of various organs including human islet β cell homeostasis [190,191,192,193]. The use of siRNA and/or specific inhibitors against selected m6A enzymes demonstrates that these enzymes modulate the expression of genes involved in pancreatic β cell identity and GSIS [192]. Increased expression of m6A methylation upregulates the IGF-1/AKT/PDX-1 pathway in human β cells, which ultimately inhibits cell-cycle arrest and protects insulin secretion [190]. PDX-1 is a key driver of β cell proliferation [194,195,196]. Wang et al. [191] found that RNA methyltransferase-like 3/14 (METTL3/14) specifically regulate both functional maturation and mass expansion of neonatal β cells before weaning. Notably, METTL3/14 directly regulates *MAFA* expression at least partially via modulating its mRNA stability. Failure to maintain this modification impacts the ability to fulfill β cell functional maturity [191]. Depletion of m6A levels in EndoC-βH1 induces cell-cycle arrest and impairs insulin secretion by decreasing AKT phosphorylation and PDX-1 protein levels [190].

Fat mass- and obesity-associated gene (*FTO*), a m6A RNA demethylase that was first identified as a susceptibility gene for obesity and T2DM, can remove m6A methylation marks from RNAs in the nuclear speckles [197]. Increased postnatal FTO expression via erasing m6A marks may compromise postnatal PDX-1-mediated β cell proliferation.

FTO-deficient Min6 cells exhibited decreased levels of *INS1* mRNA [192]. In accordance, mRNA silencing of *FTO* in GRINCH (Glucose-Responsive Insulin-secreting C-peptide-modified Human proinsulin) clonal rat β cells showed a significant reduction in GSIS [198]. It has been shown in INS-1 cells that were stably transfected with FTO-HA (hemagglutinin) that modestly increased expression of FTO selectively enhances the first phase of insulin secretion when INS-1 monolayers or pseudo-islets were stimulated with 20 mM glucose, whereas the second phase remained unchanged [199]. Furthermore, T2DM islets, which display de-differentiated β cells, revealed a reduced expression of *FTO* mRNA [199].

During the breastfeeding period characterized by increased β cell proliferation but functional immaturity of β cells with reduced GSIS, MEX may attenuate the expression of FTO. In fact, it has recently been demonstrated that miR-30b directly targets *FTO* mRNA and regulates FTO-mediated m6A methylation levels [200] (Figure 5).

Notably, miR-30b is a major miR detected in human and bovine MEX [129,201]. Further predicted miRs targeting *FTO* are miR-21, miR-22, and miR-155 [202], which are all dominant miRs of MEX. Thus, MEX-mediated FTO suppression during breastfeeding may support m6A-dependent β cell proliferation while suppressing GISIS [190].

SNPs in the second intron of the human insulin-like growth factor 2 mRNA-binding protein 2 (*IGF2BP2/IMP2*) gene are associated with an increased risk for T2DM [203]. Of importance, IMP2 is an RNA m6A reader, which promotes pancreatic β cell proliferation by enhancing PDX-1 expression [204]. IMP2 directly binds to *PDX1* mRNA and stimulates its translation in an m6A dependent manner. Moreover, IMP2 orchestrates IGF2/AKT/GSK3β/PDX-1 signaling to stable PDX-1 polypeptides [204]. Notably, *IMP2* is a target gene of let-7 [205], which is quenched by high H19 expression during the period of postnatal β cell proliferation [93], thus linking the upregulation of H19 and IMP2 during the postnatal period of β cell proliferation.

Of note is that IMP2 is the only IMP expressed in β cells and insulin-sensitive tissues in postnatal life [206]. The phosphorylation of IMP2 by mTOR and mTORC1 is critical for post-transcriptional gene expression regulation to coordinate cellular function and nutrient metabolism [206,207,208].

MEX-miR-mediated suppression of FTO expression during breastfeeding may thus contribute to IMP2/m6A *PDX1* mRNA-regulated β cell proliferation as well as FTO-dependent attenuation of GSIS (Figure 5).

## 4. MEX miR Signaling and Functional Immaturity of β Cells

Whereas neonatal β cells are highly proliferative [209], they are functionally immature, as defined by a lower set point for GSIS. Puri et al. [100] showed that β cell proliferation and immaturity are inversely linked by tuning expression of physiologically relevant, non-oncogenic levels of the transcription factor c-MYC, which lies at the nexus of most, if not all, known proliferative pathways and plays a key role for β cell proliferation [101]. Neonatal β-cells are immature and unable to secrete insulin appropriately in response to a glucose challenge [210].

### 4.1. Pyruvate Kinase

Recent evidence substantiates that pyruvate kinase (PK) controls signal strength in the insulin secretory pathway [211]. PK, which converts ADP and phosphoenolpyruvate (PEP) into ATP and pyruvate, underlies β cell sensing of both glycolytic and mitochondrial fuels. Plasma membrane-localized PK is sufficient to close K_ATP_ channels and initiate a calcium influx. The findings of Lewandowski et al. [211] support a compartmentalized model of β cell metabolism in which PK locally generates the ATP/ADP required for insulin secretion, in which PK controls the signal strength in the insulin secretory pathway. PK directly controls the “on-off” switch for insulin secretion by β cells and is a highly compartmentalized β cell fuel sensor. PK activation increases oscillatory frequency and amplifies insulin secretion [211].

It is known that c-MYC is required for the glucose-mediated induction of metabolic enzyme genes [212]. Notably, c-MYC is critically involved in the expression of glucose-responsive PK required for the PK-controlled glucose responses in insulin-secreting cells [212]. Glucose facilitates the formation of c-MYC/MAX heterodimers and increases c-MYC and MAX binding to the *PK* promoter [213]. The relative abundance of MAD, c-MYC, MAX, CHREBP, and MLX is important in the regulation of glucose-responsive genes with a strong link between c-MYC/MAX heterodimerization, inducing transcriptional activation [178]. MEX miR-148a-mediated suppression of *MAX* and thus reduced c-MYC/MAX-mediated promoter activation of PK may thus attenuate PK-controlled GSIS during breastfeeding, whereas increased PK-dependent insulin secretion after weaning could result from fading of MEX miR-148a.

### 4.2. AMP-Activated Protein Kinase

AMP-activated protein kinase (AMPK) signaling was predicted to be activated in 2-week rat islets [214]. AMPK is a master metabolic regulator controlling glucose and lipid metabolism. AMPK also plays a critical role in maintaining β cell identity. Loss of AMPK in β cells upregulates many β cell disallowed genes that are used for housekeeping functions in other tissues but are selectively repressed in islets. Disallowed genes include *SLC16A1*, *LDHA*, *MGST1,* and *PDGFRA* [215].

The glucose-sensing behavior in pancreatic β cells is also dependent on ATP-sensitive K^+^ channel (K_ATP_) activity, which is controlled by the relative levels of the K_ATP_ ligands ATP and ADP, responsible for closing and opening K_ATP_, respectively. Recent work demonstrated a critical role for AMPK in the glucose-sensing behavior of cells [216,217,218,219]. Aldolases promote the formation of lysosomal complexes containing the v-ATPase, Ragulator, AXIN, LKB1, and AMPK, previously shown to be required for AMPK activation [220,221,222]. Electrophysiological recordings, coupled with measurements of gene and protein expression in rat insulinoma cells were performed to investigate whether AMPK modulates glucose-sensing in insulin-secreting cells by altering phosphotransfer to K_ATP_ channels [223]. It has been shown in murine β cells that loss of the AMPK α2 subunit impairs GSIS and inhibits their sensitivity to hypoglycemia [224]. β cells lacking AMPK α2 or expressing a kinase-inactivated AMPK α2 failed to hyperpolarize in response to low glucose, although the K_ATP_ channel function was intact. Thus, AMPK activity is necessary to maintain normal pancreatic β cell glucose sensing. AMPK-deleted islets in vitro exhibit increased insulin release at basal glucose and decreased GSIS [224]. It has also been suggested that AMPK modifies β cell K_ATP_ activity by phosphorylation of KIR6.2 [225]. AMPK also phosphorylates Raptor [226] and TSC2 [227], thereby inhibiting mTORC1 [226,227,228]. Notably, the lysosome is a crucial hub for both AMPK and mTORC1 signaling [229].

Jaafar et al. [230] reported that the control of cellular signaling in β cells fundamentally changed after weaning and switched from the nutrient sensor mTORC1 to the energy sensor AMPK, which was critical for functional β cell maturation, mitochondrial biogenesis, and GSIS [230]. Furthermore, it was realized that mTORC1 maintains the immature phenotype of pancreatic β cells [230]. In accordance, it has recently been hypothesized that loss of MEX miRs during weaning is a potential mode of action for switching mTORC1 signaling to AMPK signaling, attenuating β cell proliferation and increasing functional β cell maturity with increased GSIS [36].

Remarkably, miR-148a targets *PRKAA1* [231], the catalytic subunit α1 of AMPK and *PRKAG2* [232], the regulatory subunit γ2 of AMPK. AMPK functions as a central mediator of mitochondrial homeostasis, orchestrating cellular responses to energetic stress and mitochondrial insults and coordinating multiple features of autophagy and mitochondrial biology [233,234].

### 4.3. MiR-375

An abundant miR of human milk and early term milk is miR-375 [15,182], which is also a major component of human and bovine MEX [201,235]. In fact, Yun et al. [236] recently confirmed that miR-375 belongs to the most abundant miRs of mature cow milk. Reif et al. [235] observed increased intestinal expression of miR-375 in mice after exposure to human and bovine MEX [235]. MiR-375 directly targets phosphoinositide-dependent protein kinase-1 (*PDPK1*) and reduces its protein level, resulting in decreased glucose-stimulatory action on insulin gene expression and DNA synthesis [237].

Notably, an overexpression of miR-375 in rat and human islet cells blunted insulin secretion in response to glucose [238]. MiR-375 enhanced lactate production, suggesting that glucose-derived pyruvate is shifted away from mitochondria. In fact, forced miR-375 expression in rat or human islets increased mRNA levels of pyruvate dehydrogenase kinase-4, but decreased those of pyruvate carboxylase and malate dehydrogenase 1 [238]. In contrast, reduced expression of miR-375 was associated with maturation of fetal rat β cells and acquisition of GSIS.

Thus, MEX-derived miR-375 may impair GSIS by redirecting glucose carbons from mitochondrial metabolism to lactate formation, thus inhibiting glucose-induced ATP generation, which is consistent with the glycolysis-dominated neonatal phenotype of β cells.

### 4.4. PPARGC1A and CPT1A

Furthermore, peroxisome proliferator-activated receptor-γ coactivator 1α (*PPARGC1A*), which is a key coregulatory transcription factor of mitochondrial function [239,240], is a target of miR-148a [241,242]. Factors responsible for mitochondrial biogenesis and degradation play key roles in the balance of mitochondrial mass in β cells. Reduced *PPARGC1A* expression in human islets by RNA interference leads to decreased insulin expression and secretion, suggesting that its expression is necessary for β cell function [243,244]. In fact, downregulation of *PPARGC1A* expression in human islets by siRNA reduced insulin secretion by 41% [244]. Moreover, miR-148a targets carnitine palmitoyltransferase 1 (*CPT1A*) [245], a central regulator of mitochondrial fatty acid β-oxidation [246,247].

Thus, MEX miR-148a-mediated suppression of *PPARGC1A* and *CPT1A* in β cells may impair mitochondrial function and ATP-dependent insulin secretion, whereas the termination of MEX signaling may enhance mitochondrial function, insulin secretion, and β cell maturation.

### 4.5. Estrogen-Related Receptor-γ

Yoshihara et al. [248] identified a novel regulator governing functional β cell maturation, i.e., the orphan receptor estrogen-related receptor γ (ERRγ; *ESRRG*) [248]. ERRγ expression is a hallmark of adult, but not neonatal, β cells. Postnatal induction of ERRγ drives a transcriptional network activating mitochondrial oxidative phosphorylation, the electron transport chain, and the ATP production needed to drive glucose-responsive insulin secretion. Mice deficient in β cell-specific ERRγ expression are glucose intolerant and fail to secrete insulin in response to a glucose challenge [248]. Remarkably, ERRγ shifts β cell metabolism from glycolysis to oxidative phosphorylation and enhances GSIS in iPSC-derived β-like cells [248]. ERRγ promotes an oxidative switch, which is regarded as a key signal for functional β cell maturation and glucose responsiveness [248,249]. Notably, orphan members of the superfamily of nuclear receptors ERRα, ERRβ, and ERRγ are of key importance for the control of mitochondrial gene networks [250,251]. ERRγ upregulates several mitochondrial genes (*MDH1*, *COX6A2*, *ATP2A2*, *NDUFS2*, and *ATP6V0A2*) [252]. Furthermore, ERRα and ERRγ stimulate the expression of pyruvate dehydrogenase kinase isoform 4 gene (*PDK4*) [253], which via inhibiting the pyruvate dehydrogenase complex switches glucose catabolism to fatty acid utilization [254], supporting β cell mitochondrial respiration under conditions of oxidative stress [255]. Of importance, ERRγ binds to its coactivator peroxisome proliferator-activated receptor-γ co-activator 1α (PGC-1α; *PPARGC1A*), producing a stable transcription factor ERRγ/PGC-1α complex [256,257]. Intriguingly, both mRNAs of *ESRRG* and *PPARGC1A* exhibit highly conserved binding sites for miR-148a [242,258]. As previously mentioned, miR-148a also suppresses *PRKAA1* [234], the catalytic subunit α1 of AMPK, and *PRKAG2* [235], the regulatory subunit γ2 of AMPK.

Thus, miR-148a-mediated suppression of *PRKAA1*, *PRKAG2*, *ESRRG,* and *PPARGC1A* functions as a synergistic inhibitory network affecting the central regulators of mitochondrial function and oxidative phosphorylation. Notably, the loss of AMPK from β cells upregulates the β cell-disallowed genes, resulting in β cell de-differentiation characterized by an increased expression of normally repressed genes, such as *LDHA*, *SLC16A1*, *MGST1*, and *PDGFRA* involved in the metabolic pathways of glycolysis [215].

Notably, hypoxia-inducible factor-1α (HIF-1α) was identified as a key regulator of hypoxia-induced de-differentiation of β cells by upregulating mature β cell-disallowed genes including *LDHA* [259], discussed later in more detail. ERRα and ERRγ are regarded as key players of metabolic control [260]. In breast cancer cells, ERRγ has been shown to be a negative regulator of anaerobic glycolysis and inhibition of ERRγ by miR-378 * led to an increase in lactate production and a decrease in aerobic respiration [261]. Furthermore, siRNA of ERRγ led to an increase in LDHA expression [261].

It is thus conceivable that MEX miR-148a-mediated suppression ERRγ may also maintain β cell anaerobic glycolysis during the breastfeeding period, whereas weaning relieves ERRγ suppression. To understand the molecular role of ERRγ and elucidate the potential key candidate genes involved in pancreatic β cells, an eukaryotic expression plasmid containing mouse ERRγ was constructed and transfected into NIT-1 pancreatic β cells. Overexpression of ERRγ in pancreatic β cells enables regulation of the expression of certain genes involved in cell apoptosis and mitochondrial function, such as *TFPT*, *BCL7C*, *DAP*, *THOC6*, *UBE2D3*, *ATP5H*, *MPV17*, and *NDUFA6*, respectively [262].

Upregulation of ERRγ expression after weaning—the physiological termination of MEX miR-148a signaling—may also upregulate TGF-β signaling, promoting functional maturation of β cells. It has been shown that ERRγ directly binds to an ERR response element in the *TGFB2* promoter to induce TGF-β2 transcription [263]. Furthermore, ERRγ has also been identified as a key regulator of hepatic gluconeogenesis [264]. An inverse agonist of nuclear receptor ERRγ suggested for the reduction of hepatic gluconeogenesis [265] may thus deteriorate ERRγ-dependent β cell homeostasis.

Although the process that induces activation of ERRγ expression after weaning is not fully understood [249], translational evidence indicates that fading of MEX miR-148a signaling promotes the oxidative switch to functional β cells (Figure 6).

ERRγ may also affect cell cycle progression, as demonstrated in prostate cancer cells where ERRγ induced the expression of cell cycle inhibitors p21 and p27, suppressing cell proliferation [266].

### 4.6. DNA Methyltransferase 3A

Dhawan et al. [267] provided fundamental evidence that during postnatal life, the de novo DNA methyltransferase 3A (DNMT3A) initiates a metabolic program by repressing key genes, thereby enabling the coupling of insulin secretion to glucose levels. Notably, β cell-specific deletion of DNMT3A prevented the metabolic switch, resulting in loss of GSIS. DNMT3A binds to the promoters of hexokinase 1 (*HK1*) and *LDHA*, key genes involved in the regulation of the metabolic switch from glycolysis to oxidative phosphorylation. Knockdown of these two key DNMT3A targets restored the GSIS response in islets from animals with β cell-specific DNMT3A deletion. In addition, DNA methylation-mediated repression of glucose-secretion decoupling genes to modulate GSIS was conserved in human β cells. Thus, epigenetic regulation by DNA methylation plays a central role in shaping the gene expression patterns that define the fully functional β cell phenotype and regulate β cell growth [268]. The specificity of DNA methylation patterning is ensured by the interaction of DNMT3A with transcription factors NKX2.2 and NKX6.1, allowing recruitment to specific sites. Once established, the β cell specific DNA methylation patterns are maintained by the maintenance methyltransferase DNMT1 [268]. It has been known for a decade that ERRγ functions as a transcriptional activator of mouse and human DNMT1 expression by direct binding to its response elements (ERE1/ERE2) in the *DNMT1* promoters [269]. Indirect evidence in mice fed a high-fat diet supports the view that ERRγ upregulates the expression of DNMT3A [270]. In fact, the partial ERRγ antagonist γ-oryzanol decreased the mRNA and protein expression of DNMT1, DNMT3A, and DNMT3B [270]. Thus, increased ERRγ expression at the time of weaning associated with a loss of MEX miR-148a may initiate the DNMT3A-mediated epigenetic switch to functionally mature β cells (Figure 6).

Recent evidence observed in mouse embryonic stem cells demonstrates that DNMT1 regulates the timing of DNA methylation by DNMT3 in an enzymatic activity-dependent manner [271]. The termination of MEX miR-148a signaling during weaning may enhance the expression of miR-148a targets DNMT1 and ERRγ, promoting the epigenetic switch to mature β cells. In addition, the disappearance of MEX miR-29b, [272,273,274], MEX miR-30 [129,201], and MEX miR-22 [182], which all target *DNMT3A* [275,276,277], may further enhance DNMT3A expression after weaning.

To understand the regulatory impact of ERRγ in β cell maturation and DNMT3A expression, it should be considered that ERRγ binds to nuclear receptor interacting protein 140 (RIP140), with the receptor in the agonist conformation setting up a situation in cells where there is a dynamic competition between RIP140 and coactivators such as nuclear receptor coactivator 1 (NCOA1) and PGC-1a [278]. Weaning-associated decline of MEX miR-30b, which targets RIP140 (*NRIP1*) [279], may thus additionally modify the transcriptional regulation of ERRγ.

### 4.7. β Cell Glycolysis: Impact of p53, TIGAR, SCO2, and HIF-1α

The glycolytic flux plays a key role in β cell proliferation, providing a direct link between HIF-1α and the maintenance of islet mass [280]. Recent evidence indicates that p53 balances glycolysis and aerobic respiration [281].

MEX miR-148a and miR-125b-mediated suppression of p53 [132,150] may attenuate p53-mediated transcriptional expression of *STK11* (LKB1) [137,142], a positive regulator activating AMPK that inhibits mTORC1 [226,227]. AMPK phosphorylates and thereby activates TSC2 [228]. The mTORC1-activating upstream genes *PTEN* and *TSC2* are target genes of p53 [137,138,139,140,141]. Furthermore, *TSC2* is a predicted target of miR-148a [282]. Thus, p53, at multiple checkpoints, suppresses mTORC1 [139], the key driver promoting c-MYC [174] and HIF-1α expression [283].

A critical downstream effector of glycolysis is TP53-induced glycolysis and apoptosis regulator (*TIGAR)* [284]. *TIGAR* as a p53-inducible gene that lowers fructose-2,6-bisphosphate levels in cells, resulting in an inhibition of glycolysis [284]. Of importance, p53 not only regulates glycolysis but also oxidative phosphorylation (OXPHOS) via direct promotion of SCO cytochrome c oxidase assembly protein 2 (*SCO2*) expression [285].

SCO2 operates at the inner membrane of the mitochondria where it facilitates the assembly of cytochrome c oxidase complex in the electron transport chain. Small interfering RNA (siRNA)-mediated suppression of p53 in HCT116 cells decreased SCO2 protein expression and reduced oxygen consumption corresponding to p53-deficient cells [285]. Thus, p53 antagonistically regulates the inter-dependent glycolytic and OXPHOS cycles [286]. A further putative mechanism, whereby p53 inactivation promotes glycolysis, may include increases in hexokinase 2 (*HK2*) and phosphoglycerate mutase (*PGM*) [281].

It is thus conceivable that MEX miR-148a/miR-125b via lactation-dependent inhibition of p53 supports glycolysis but attenuates SCO2-dependent OXPHOS, explaining the metabolic switch from glycolysis and OXPHOS-dependent β cell maturation after weaning (Figure 7).

Van de Velde et al. [287] demonstrated that glucagon-like peptide 1 (GLP-1) stimulates HIF-1α accumulation via the cAMP-mediated induction of the mTORC1 pathway in β cells. cAMP appears to trigger mTORC1 activation in part via the CREB-mediated induction of IRS2-AKT signaling. Thus, mTORC1 links incretin signaling to HIF-1α induction in pancreatic β cells [287]. HIF-1α is a key transcription factor activating glycolytic enzymes [288] and plays a crucial role in the regulation of glycolytic enzymes in β cells [289]. As far as we know, β cell glycolysis resembles Warburg-type glucose metabolism, which has also been described in early mammalian embryos to support rapid cell proliferation and growth [290,291,292,293,294]. Notably, HIF-1α transcriptional activity is controlled by factor inhibiting HIF-1 (FIH-1), a protein that binds to von Hippel-Lindau tumor suppressor protein (VHL) and functions as a transcriptional co-repressor inhibiting HIF-1α transactivation and function [295]. Intriguingly, *HIF1AN*, the gene encoding FIH-1, is a predicted target gene of miR-148a [296]. Thus, MEX miR-148a may enhance HIF-1α-induced β cell glycolysis, which correlates with β cell de-differentiation and upregulation of disallowed genes [259].

HIF-1α promotes aerobic glycolysis by upregulating the expression of pyruvate dehydrogenase kinase (PDK), which inhibits pyruvate dehydrogenase (PDH), the enzyme converting pyruvate to acetyl-CoA [297,298]. p53 promotes mitochondrial respiration while inhibiting glycolysis [299] (Figure 7).

p53 via upregulation of TIGAR decreases the activity of 6-phoshofructo-1 kinase (PFK-1) [300,301], which controls the commitment of glucose to glycolysis via irreversible conversion of fructose-6-phosphate to fructose-1,6-biphosphate. MEX miR-mediated suppression of p53 may thus enhance PFK-1 expression in β cells promoting glyoslysis and attenuating mitochondrial respiration. Notably, PFK-1 and its isoforms play a critical role in the regulation of β cell glycolysis [302,303,304,305]. Recently, Haythorne et al. [306] reported significant upregulation of multiple genes of glycolysis-related genes including PFK-1 in islets of diabetic βV59M mice.

Weaning, the physiological termination of MEX miR signaling, may thus shift β cell metabolism from mTORC1/c-MYC/HIF-1α-dominated β cell proliferation/glycolysis to AMPK/OXPHOS-dependent maturity and GSIS. Intriguingly, significantly increased levels of miR-148a have been observed in the serum of patients with T2DM [176]. MiR-148a may thus represent the key driver promoting β cell de-differentiation back to the postnatal proliferative/glycolytic phenotype [36]. Furthermore, it is already well appreciated that miRs are critically involved in metabolic reprogramming, favoring glycolysis, thereby supporting biosynthetic pathways required for cell proliferation [307] (Figure 7).

### 4.8. MAFB

After birth, a significant decline of MAF bZIP transcription factor B (MAFB) has been observed, whereas MAFA continuously increases [308]. The transition from MAFB to MAFA is critical to β cell function [308,309,310,311,312]. MAFA is solely required for GSIS in adult islet β cells and is not involved in β cell islet cell development [309,312]. We note that *MAFB* is a direct target of miR-148a [313] and miR-155 [314], respectively.

Gao et al. [46] studied a crosstalk between ATM-derived exosomes (ATM-EX) and β cells (Figure 1). Exosome transfer of miR-155 to β cells repressed *MAFB,* resulting in increased numbers of proliferating β cells. In addition, knockdown of *MAFB* blunted insulin synthesis and GSIS in both human and mouse islets, accompanied by a significant repression of GLUT2 (*SLC2A2*) [46], a MAFB-regulated gene [314]. In accordance, GLUT1 expression, which is also activated by MAFB [315], is required for the formation of glucose-responsive β cells [316]. Remarkably, GLUT1 (*SLC2A1*) is also a predicted target gene of miR-148a [317].

### 4.9. ABCA1

β cell deficiency for the ATP binding cassette transporter A1 (ABCA1), which mediates the efflux of cellular cholesterol, leads to altered intracellular cholesterol homeostasis and impaired insulin secretion in mice [318,319]. Carriers of loss-of-function mutations in *ABCA1*, including patients with Tangier disease [320], show impaired insulin secretion without insulin resistance [321,322]. Lack of β cell ABCA1 leads to impaired exocytosis of insulin granules [323]. *ABCA1* mRNA is also a direct target of miR-148a and miR-106b [324], which are exosomal miRs detected in the mature milk of Holstein cows [325].

### 4.10. TGF-β and FoxO1 Signaling

TGF-β signaling plays important roles during endocrine pancreas development, β cell proliferation, differentiation, and apoptosis [326,327,328]. Recent evidence indicates that the suppression of TGF-β receptor signaling in β cells increase β cell proliferation in mice [329]. Treatment of human islets with a TGF-β receptor 1 inhibitor, SB-431542 (SB), significantly increased the β cell number by increasing β cell proliferation. SB-mediated suppression of TGF-β receptor 1 increased cell-cycle activators and decreased cell-cycle suppressors in human β cells [329]. In accordance, increased β cell mass and proliferation has been observed in β cell-specific SMAD2-deficient mice [330]. Dhawan et al. [331] showed that TGF-β signaling via SMAD3 integrates with the trithorax complex to activate and maintain INK4A (*CDKN2A*) expression to prevent β cell replication. Importantly, inhibition of TGF-β signaling repressed the INK4A/ARF locus, resulting in increased β cell replication in adult mice [331]. Furthermore, small molecule inhibitors of the TGF-β pathway promoted β cell replication in human islets transplanted into NOD-scid IL-2Rg(null) mice. In accordance with these findings, SMAD3 deficiency completely protected against diabetes-associated β cell loss and dysfunction in db/db mice. SMAD3 deficiency preserved the expression of β cell development mediator PAX6 in islets, thereby enhancing β cell proliferation and function in db/db mice in vivo and in Min6 cells in vitro [332].

In contrast, increased levels of TGF-β1 ligand and phosphorylation levels of TGF-β’s chief transcription factor, SMAD3, in human islets, promoted β cell apoptosis in human T2DM islets [333]. SMAD3 phosphorylation is similarly increased in diabetic mouse islets undergoing β cell apoptosis and loss of β cell mass [333]. SMAD3 associates with FOXO1 to propagate TGF-β-dependent β cell apoptosis. Indeed, genetic or pharmacologic inhibition of TGF-β/SMAD3 signals or knockdown of *FOXO1A* protects β cells from apoptosis [333].

Notably, miR-148a-mediated inhibition of TGF-β/SMAD2 signaling has been shown in hepatocellular carcinoma cell lines [334]. *SMAD2* mRNA is a direct target of miR-148a [334]. Another important miR-148a target gene is ubiquitin-specific protease 4 (*USP4*) [335,336,337]. Loss of UPS4 leads to the activation of several p53-directed pathways and attenuates TGFBR1 activity [338,339,340]. USP4 can de-ubiquitinate TGF-β type 1 receptor and sustain its plasma membrane expression [338,340].

MEX miR-148a-mediated suppression of *USP4* and *SMAD2* and miR-148a-mediated suppression of *TP53* and its target gene *FOXO1A* [137], a negative coregulator of SMAD3 [333], may thus attenuate β cell TGF-β/SMAD2/SMAD3 signaling promoting β cell proliferation and mass expansion (Figure 8).

The termination of MEX miR-148a signaling after weaning may shift TGF-β signaling to higher magnitudes. In fact, TGF-β stimulates insulin gene transcription, insulin secretion and islet function, and blocks mitogenic responses of pancreatic β cells to glucose [341,342]. Thus, MEX miR-148a maintains postnatal β cell proliferation for β cell mass expansion, whereas fading of MEX miR-148a signals apparently promotes TGF-β-mediated functional β cell maturation and differentiation. Weaning of milk and MEX miR signals with consecutive reduction of β cell H19 expression and increase in miR-29b may further enhance TGF-β signaling and β cell differentiation [124].

### 4.11. NEUROD1 and NKX2.2

The terminal definitive differentiation into β mature cell identity is achieved by the final activation of pivotal transcription factors, such as NEUROD1, NKX2.2, NKX6, PAX4/6, PDX-1, MAFA, FOXA2, and MNX1 [343,344]. NEUROD1 (also known as BETA2) is a basic helix-loop-helix (bHLH) transcription factor, which is crucial for pancreatic development and is essential for the maturation and differentiation of β cells and insulin production [345]. NEUROD1, together with PDX-1, ISL1, and MAFA, are key transcription factors regulating insulin synthesis in pancreatic β cells in response to blood glucose [346]. Notably, NEUROD1 and PDX-1 act upon the insulin enhancer to control β cell-specific insulin gene transcription [346]. Remarkably, *NEUROD1* is a target gene of miR-148a and miR-30b [347].

In diabetes, endocrine identity recapitulates the less mature/less-differentiated fetal/neonatal cell type, possibly representing an adaptive mechanism. The concerted activities of PAX4 and NKX2.2 are essential to initiate pancreatic β cell differentiation [348,349]. Of importance, *NKX2.2* is a target of miR-30b [350]. It is thus conceivable that MEX miR-148a und miR-30b attenuate β cell differentiation and insulin synthesis during the postnatal period of β cell proliferation (Table 1).

## 5. Weaning: The Metabolic Switch Induced by Fading of MEX miR Signaling

Human milk contains ~2.2 × 10^11^ exosomes/mL [353]. The daily transferred number of MEX by daily uptake of 800 mL human milk is tremendous and has been approximated to be ~176 trillion [353]. Notably, when an exosome- and RNA-depleted diet was initiated in mice at weaning and continued for ~4 weeks, miR levels in plasma, liver, skeletal muscle, intestinal mucosa, and placenta decreased by up to 60% compared with controls fed an exosome and RNA-enriched diet, clearly pointing to a loss of MEX and miRs in various tissues after withdrawal of MEX [31,33,272,354,355]. Unfortunately, these studies did not pay attention to pancreatic islets. Weaning, the termination of milk and MEX miR signaling, may change miR-mediated postnatal gene repression of specific β cell genes, thereby switching the metabolic profile of the β cell from a lactation-dominated mTORC1/c-MYC-driven proliferative and immature neonatal phenotype to a differentiated AMPK/ERRγ/TGF-β-activated mature and glucose-responsive adult phenotype [36,230].

## 6. Perinatal Factors Disturbing MEX miR-Regulated β Cell Homeostasis

The perinatal period is a time of fast physiological changes dependent on epigenetic programming [13]. Adverse events may lead to epigenetic changes, with implications for health and disease [13,356,357,358,359,360,361,362,363,364,365,366,367,368]. Epigenetic alterations have been linked to early life environmental stressors, including mode of delivery, famine, psychosocial stress, severe institutional deprivation and childhood abuse [369]. Recently, several maternal and iatrogenic factors have been identified that modify MEX miR composition and thus eventually affect MEX miR signaling.

### 6.1. Preterm Birth

Human MEX miR expression patterns also respond to preterm birth conditions. Kahn et al. [182] demonstrated that the expression MEX miR-22 and miR-148a are significantly upregulated in MEX of mothers giving birth to preterm infants compared to term infants. Shiff et al. [370] confirmed that the expression of miR-148a was higher in the colostrum of preterm than in full-term human milk, whereas miR-320 was more highly expressed in the colostrum of full-term than in preterm human milk.

Preterm MEX miR composition may influence early postnatal development of the pancreatic β-cells of preterm infants. Remarkably, *MECP2* is a highly conserved target of miR-22, miR-148a, and miR-30 [371,372]. MECP2 is important for gene silencing and guides DNMT1 to CpG sides for DNA methylation [109], which generally suppresses gene transcription [184]. Thus, preterm MEX miR-22 and miR-148a via targeting *MECP2* and *DNMT1*, respectively, may suppress gene promoter methylation and thus promote gene transcription, a potential mode of action accelerating β cell growth of the preterm infants. In contrast, inhibition of miR-22 in primary cultures of human subcutaneous adipocytes resulted in increased lipid oxidation, mitochondrial activity, and energy expenditure. These effects may be mediated through activation of target genes such as *KDM3A*, *KDM6B*, *PPARA*, *PPARGC1B*, and *SIRT1*, which are involved in lipid catabolism, thermogenesis, and glucose homeostasis [373,374]. MiR-22 reduces the expression PGC-1α, the co-transcription factor of ERRγ, synergistically promoting mitochondrial biogenesis. It has been shown in hepatocytes that inhibition of miR-22 activates AMPK [375]. In muscle cells, miR-22 inhibits AMPK/SIRT1/PGC-1α signaling [376,377]. In murine models and primary brown adipocytes, miR-22 activates mTORC1 signaling by directly suppressing *TSC1* and *HIF1AN*, promoting glycolysis, and maintaining thermogenesis [378]. *HIF1AN* is also a predicted target of miR-148a [297]. The thermogenic activity of miR-22 is of critical importance for the survival of premature infants, and apparently miR-22 via suppressing AMPK and activating mTORC1 may promote β cell proliferation and glycolytic activity of the premature neonatal β cells [378]. Of note, VHL, which forms a complex with HIF1AN on the promoter of HIF-1α, represses HIF-1α transcriptional activity [295]. *VHL* mRNA is a direct target of miR-21 [379,380,381,382,383], a signature miR of human and bovine milk [15,16,17,18,19,158]. Deletion of VHL protein in β cells resulted in HIF-1α activation, leading to increased anaerobic glycolysis with impaired GSIS [384], the postnatal metabolic profile of proliferating and functionally immature β cells.

Furthermore, miR-22 targets *TP53* [385], and thus augments p53-dependent glycolysis [281,284,285], which promotes the anabolic and biosynthesis of the critical components required for cell growth [299]. Fortified MEX miR-22- and miR-148a signaling of preterm milk may thus enhance glycolysis and proliferation of β cells to compensate for the prematurity of β cells. Moreover, MEX miR-148a/miR-22-mediated suppression of *HIF1AN* may attenuate postnatal HIF-1α-regulated β cell glucose sensing [384].

Whole milk collected within month 2 of lactation from mothers of preterm infants showed stable expression of miR-16 and miR-21 [386]. Notably, miR-16-5p inhibits the apoptosis of high glucose-induced pancreatic β cells via targeting of chemokine with CXC motif ligand 10 [387]. Circulating serum levels of miR-16-5p were upregulated in women with gestational diabetes [388]. Remarkably, miR-16-5p directly targets sestrin 1 (*SESN1*) [389], whereas miR-148a-3p converges with miR-16-5p in targeting sestrin 2 (*SESN2*) [390]. The p53 target genes sestrin 1 and sestrin 2 connect genotoxic stress and mTORC1 signaling [391]. Sestrin 1 and sestrin 2 activate AMPK phosphorylating TSC2 and stimulate its GAP activity, thereby inhibiting mTORC1 [391]. Furthermore, sestrins can bind the mTORC1-regulating GATOR2 protein complex, which was postulated to control the trafficking of mTORC1 to lysosomes [392]. Sestrin 2 inhibits mTORC1 activity via GATOR regulation and inhibits mTORC1 lysosomal localization via a Rag-dependent mechanism [393]. Sestrin 2-GATOR2-GATOR1-RagB signaling mediates the stress-dependent suppression of mTORC1 activity [394]. Sestrins function as guanine nucleotide dissociation inhibitors for Rag GTPases to control mTORC1 signaling [395] and regulate the localization of mTORC1 in response to amino acids [396,397].

In premature infants, plasma insulin levels are increased by amino acid administration, but glucose infusion is ineffective in stimulating insulin release [398]. Notably, protein levels in colostrum and mature human milk are increased in mothers delivering preterm infants compared to the protein content of term infants [399,400]. Apparently, adaptations of maternal milk protein content and MEX miR composition via suppression of sestrin-AMPK signaling may accelerate compensatory mTORC1-dependent β cell growth of the preterm infant.

### 6.2. Maternal Stress during Pregnancy

Maternal stress is associated with adverse child health [401]. Bozack et al. [402] studied the associations between maternal lifetime stressors and negative events in pregnancy and MEX/EV miRs. The expression level of 8 and 17 exosome/EV miRs was associated with Life Stressor Checklist-Revised Survey (LSCR) and Negative Life Event (NLE) scores, respectively. Among the primary miRs associated with LSCR scores was miR-155, whereas miR-96 was associated with increased NLE scores, respectively [402]. Notably, *MAFB* and *PTEN* are direct targets of miR-155 [160,403]. MiR-96 directly targets *AKT1S1* [404], the gene encoding proline-rich AKT substrate 40-KD, which functions as a negative regulator of mTOR kinase [405,406,407,408,409].

### 6.3. Maternal Obesity

Maternal obesity is a major risk factor of T2DM of the offspring [410,411,412,413,414]. The first 1000 days of life have been postulated to be the key period for T2DM prevention [412]. Maternal obesity, the human milk metabolome, and human milk fat showed associations with infant body composition and postnatal weight gain [411,415,416]. Of note, maternal obesity modifies the concentrations of key MEX miRs [417]. In the overweight/obese group of women compared to the normal weight group at 1 month after birth, the abundance of MEX miR-148a and miR-30b was lower by 30% and 42%, respectively [417]. In addition, the levels of miR-30b, let-7a and miR-378 in colostrum were negatively correlated with maternal pre-pregnancy BMI, whereas in mature milk, let-7a was negatively correlated with maternal weight late in the pregnancy [418]. Reduced MEX miR-148a and miR-30b levels in the milk of obese mothers may thus negatively affect MEX miR-mediated β cell proliferation.

### 6.4. Gestational Diabetes Mellitus

Epidemiological and epigenetic evidence points to an increased risk of T2DM in the offspring of mothers with gestational diabetes mellitus (GDM) [419,420,421,422,423,424,425]. Suckling of normal Wista rats by a diabetic Goto-Kakizaki rat had a negative impact on metabolic programming of β cell of newborn rats associated with a reduction of β cell mass, resulting in long-term glucose intolerance [426]. GDM changes the metabolomes of human colostrum, transition milk, and mature milk [427,428]. Notably, Shah et al. [429] observed reduced levels of MEX miR-148a, miR-30b, let-7a, and let-7d in the milk from mothers with GDM.

### 6.5. Maternal Diet

The maternal diet has an influence on the levels of human milk miRs. Hicks et al. [430] reported that nearly half of abundant miRs were impacted by diet. Of interest, eicosapentaenoic acid via binding to free fatty acid receptor 4 (FFAR4) enhances the expression of miR-30b and miR-378 [431]. Remarkably, levels of miR-30b and miR-378 in the colostrum exhibit a negative relation to maternal pre-pregnancy BMI [418]. Preferential intake of n-3 PUFA may thus modify the m6A RNA methylation status of the β cell via miR-30b-mediated suppression of *FTO* [200]. Consumption of the “Cafeteria diet”—a standardized animal model of hypercaloric Western junk food [432]—during lactation (day 15) in rats, resulted in higher levels of miR-222 in rat milk compared to the miR-222 levels of the control animals [433,434]. Of note, serum exosomes enriched in miR-222 after bone marrow transplantation increased murine β cell proliferation in mice after STZ-induced β cell injury [435]. The cell cycle inhibitor *CDKN1B* is a conserved target gene of miR-148a and miR-222 [436].

### 6.6. Caesarean Delivery

Plasma oxytocin levels increase gradually during pregnancy and especially during labor [437]. Gutman-Ido et al. [438] found that oxytocin upregulates miR-148a and miR-30 in human colostrum but downregulates miR-320. Notably, miR-320 was highly expressed compared with miR-148a in the colostrum of mothers who did not receive exogenous oxytocin [438]. It has been shown in breast cancer cells and diabetic mice pancreatic tissue that miR-320 attenuates PI3K/AKT/ELF3 signaling [439,440]. E74-like factor 3 (*ELF3*) is a direct target of miR-320 [439] and silencing of *ELF3* has been shown to promote β cell apoptosis [441]. It is thus conceivable that caesarean section, associated with deficient oxytocin signaling compared to vaginal labor, disturbs the balance of prosurvival/proapoptotic miR-148a/miR-320 signaling, compromising β cell mass expansion, which may enhance the risk of T2DM later in life. In fact, Chiba et al. [153] recently demonstrated that that miR-148a and miR-125b are significantly reduced in transition and in the mature milk of mothers giving birth via caesarean delivery compared with MEX miR levels observed with normal vaginal birth. Furthermore, caesarean section reduces the prevalence of early breastfeeding [442] and may thus negatively affect postnatal MEX miR-mediated programming of β cells.

### 6.7. Changes of miR Levels during Breastfeeding

The concentration of miRs change during the process of lactation [153,430]. The levels of miR-148a and miR-125b were lower in mature milk compared to transition milk [153]. Between the second week and the third month of lactation of healthy mothers delivering term infants, seven human milk miRs had significant stage-specific upregulation (miR-3184-3p, miR-92b-5p, let-7d-3p, miR-516a-5p, miR-187-5p, miR-3126-5p, and miR-196a-5p), whereas four had significant stage-specific downregulation (miR-125b-5p, miR-146a-5p, miR-34a-5p, and miR-1307-3p) [443].

### 6.8. Duration of Breastfeeding

Growing evidence indicates that the duration of breastfeeding plays a crucial for postnatal epigenetic imprinting and long-term clinical outcomes [444,445,446,447,448]. Breastfeeding has been related to the modification of methylation markers associated with T2DM [449]. Observational evidence suggests that breastfeeding reduces the risk of both T1DM and T2DM [450,451,452,453]. The duration of breastfeeding may thus have an impact on the exposure time and quantity of MEX miRs and their potential epigenetic signaling effects on pancreatic islets, enhancing β cell proliferation and mass expansion.

### 6.9. Human Donor Milk

Human donor milk is an important source of milk, especially for preterm infants whose mothers are unable to provide milk [454,455]. Perri et al. [456] reported that the method of human milk Holder pasteurization (HoP) of 62.5 °C, 30 min, did not alter the distribution or the expression profile of four selected miRs (miR-21, miR-181a, miR-150, and miR-223) tested in both colostrum and human milk. Smyczynska et al. [457] confirmed miR-148a in the greatest amount, accounting for almost 24% of total exosomal miRNA and about 12% in whole milk, prior to Holder pasteurization (HoP) and high pressure processing (HPP) 450 MPa for 15 min. HoP led to a 82-fold decrease in whole RNAs and a 302-fold decrease in exosomes [457]. After HPP, the percentage of miR-148a dropped to about 1/3 of its level in raw milk, whereas pasteurization affected miR-148a recovery to a much higher degree. Thus, in comparison to physiological breastfeeding, the application of HoP-processed human donor milk is associated with a critical loss of exosomal miR-148a, which may adversely affect β cell proliferation.

### 6.10. MiR-Deficient Artificial Formula Feeding

Compared to raw cow milk, significant reductions of miR levels have been measured in powdered formula [158]. Leiferman et al. [458] reported that milk miRs were not detectable in infant formulas. The levels of miR-148a and miR-125b in all recently analyzed infant formulae were lower than 1/500^th^ and 1/100^th^ of those in mature human milk, respectively [153]. Lectins in soy formula bind bovine MEX and prevent their absorption in healthy adults [459]. Thus, breastfeeding combined with soy-based complementary formula may impair the intestinal MEX miR uptake of the infant, which is of concern, as soy-based formula has been advocated as a substitute for infants with cow milk allergy [460,461]. Soy formula may thus interfere with MEX uptake and MEX miR signaling, which are important for allergy prevention [19,462,463]. Of note, miR-148a via suppression of *DNMT1* controls the expression of *FOXP3*, the master transcription factor of regulatory T-cells, which are reduced in atopic infants [462]. Furthermore, miR-375 binds directly to *JAK2* mRNA, reducing its expression [464,465,466,467]. Increased JAK2 signaling has been associated with the pathogenesis of atopic diseases [468,469,470,471]. Of note, JAK1/JAK2 inhibition reversed established autoimmune insulitis in NOD mice [472]. The first signs of β cell autoimmunity may be initiated during the first year of life, implying that risk factors for β cell autoimmunity and T1DM must be operative in early infancy [449,450,451,452]. An adequate miR-375 gene dosage in pancreatic β cells plays an essential role in the maintenance of β cell mass [473].

Cheshmeh et al. [474] reported that the expression levels of *FTO* and *CPT1A* genes in mononuclear blood cells of formula-fed and mix-fed infants was significantly higher compared to the exclusive breastfeeding group. As far as we know, MEX miR-deficient formula is unable to provide sufficient miR-30b and miR-148a to target *FTO* and *CPT1A* expression, respectively, which are critically involved in m6A-dependent β cell proliferation and CPT1A-mediated mitochondrial fatty acid oxidation. Thus, formula feeding may bear the risk of the deviated epigenetic regulation required for appropriate β cell proliferation [190,191]. Table 2 summarizes the perinatal factors that modify MEX miR expression.

## 7. Bovine MEX miR Signaling during Adult Human Life

In the time scale of human evolution, the consumption of cow milk is a novel behavioral change of humans [477], modifying human physiology, such as accelerated linear growth [478]. Cow milk signaling affects a large spectrum of tissues in the human body likely including the pancreatic islets [479]. Western societies are exposed to substantial daily amounts of pasteurized cow milk [480,481,482,483]. Pasteurized milk is consumed in high quantities in the United States and northern European countries. Swedes, for instance, who per capita consumed 98 L milk in 2018, primarily drink pasteurized milk [484]. Pasteurization (72 °C for 15 s) is a gentle thermal processing in contrast to ultraheat treatment (UHT) (135–150 °C, 1–10 s) [485]. Unfortunately, no study so far has investigated the impact of thermal processing, especially pasteurized versus UHT milk intake and risk of T2DM [484]. This is of critical importance because consumers of pasteurized milk are exposed to bioactive MEX, while MEX miRs are significantly reduced by UHT [475,476]. Kleinjan et al. [475] showed that UHT of milk resulted in a loss of EVs, whereas EV numbers after pasteurization were not affected. Although pasteurization and homogenization of commercial milk reduced total milk-EV-associated RNAs (from 40.2 ± 3.4 ng/μL in raw milk to 17.7 ± 5.4–23.3 ± 10.0 ng/μL in processed milk), miR-148a, miR-21, and miR-30d, although reduced, were still detectable in pasteurized and homogenized milk [475]. Although pasteurization of cow milk reduces the concentration of milk EVs and MEX, bovine milk contains 10^12^–10^14^ exosomes per mL [353] and thus delivers 10- to 1000-fold higher MEX numbers/mL compared to human milk. Zhang et al. [476] recently confirmed that bioactive miRs including miR-9 in raw milk were lost after UHT but not after pasteurization. Mutai et al. [486] confirmed that miR-21 is a component of MEX derived from pasteurized commercial cow milk. Humans and cows share identical miR-21-5p sequences [487]. Importantly, miR-21 plasma levels significantly increased by 147% 3.2 h in human volunteers after consumption of 1 L commercial milk [486]. MiR-21, an abundant signature miR of cow milk [15,158], and miR-9 both target *FOXO1A* [488,489,490,491,492]. MiR-21 orchestrates high glucose-induced signals to mTORC1, resulting in renal cell pathology in diabetes [493]. Furthermore, miR-21 is upregulated in the adipose tissue of obese diabetic subjects [494]. In fact, miR-21 expression was two-fold higher in the adipose tissue in patients with T2DM. In primary cultures of adipocytes from non-diabetic overweight subjects, miR-21 expression increased after 24-h exposure to high glucose and insulin [494]. Circulating miR-148a-5p and miR-21-5p have been identified as novel diagnostic biomarkers in adult male patients with metabolic syndrome [495]. MiR-148a and miR-21 in a synergistic fashion decrease FoxO1 nuclear activity and attenuate TGF-β signaling. MEX miR-148a targets p53 [132], the key transcription factor maintaining FoxO1 baseline expression [137] (Figure 8).

Talchai et al. [496] provided evidence that not β cell death—but β cell de-differentiation—is the primary mechanism of diabetic β cell failure. In this regard, it is of pivotal importance that FoxO1 integrates β cell proliferation with adaptive β cell function. In mice lacking FoxO1 in β cells, FoxO1 ablation caused hyperglycemia with reduced β cell mass, following physiologic stress such as multiparity and aging [497]. Lineage-tracing experiments demonstrated that loss of β cell mass was due to β cell de-differentiation, but not cell death [497]. Notably, de-differentiated β cells reverted to progenitor-like cells expressing NEUROGENIN3, OCT4, NANOG, and L-MYC observed in postnatal β cells during the breastfeeding period [7]. In fact, overexpression of miR-21 increased proliferation but decreased net β cell numbers [497]. Recently, Ibrahim et al. [498] demonstrated that β cell pre-miR-21 induces dysfunction and loss of cellular identity by targeting *TGFB2* and *SMAD2* mRNAs. Thus, miR-21 signaling converges with miR-148a signaling in suppressing FoxO1 and TGF-β signaling, key regulators maintaining β-cell differentiation (Figure 8).

Thus, continued consumption of pasteurized cow milk with persistent MEX miR signaling after the physiological weaning period appears to be a critical change of human physiology that may promote β cell de-differentiation back to the proliferative/glycolytic postnatal phenotype characterized by increased miR-148a/miR-21/mTORC1 and decreased AMPK and TGF-β signaling [36,230].

### 7.1. Type 1 Diabetes Mellitus

Although prospective birth cohort studies have not shown any link between early exposure to cow milk and T1DM [499,500], recent evidence indicates that the early introduction of cow milk proteins is a risk factor in the pathogenesis of T1DM [501]. Short duration and/or a lack of breastfeeding may constitute a risk factor for the development of T1DM later in life [502], whereas cow milk consumption later in childhood has showed contradictory results [503,504,505,506]. Remarkably, a systematic review and bioinformatic analysis demonstrated that circulatory levels of miR-148a-3p, miR-21-5p, and miR-375 among others were significantly upregulated in patients with T1DM compared to controls [507]. Recent findings suggest that exosomes and their cargo are associated with the development of T1DM [41,53,508].

Bovine MEX are taken up by human macrophages [201] as well as murine bone marrow-derived macrophages [509]. The communication between islet macrophages and β cells is the recent focus of T1DM and T2DM pathogenesis [510]. Notably, miR-148a via targeting *MAFB* facilitates inflammatory monocyte-derived dendritic cell differentiation and autoimmunity [511], which is involved in the immune pathogenesis of T1DM [512,513]. Macrophages are observed in islets of patients with recent onset T1DM and are the first cells identified in islets of animal models of T1DM. T1DM is prevented in non-obese diabetic mice that have been depleted of macrophages [514]. MiR-148a promotes proinflammatory M1 macrophage polarization [515], a critical finding observed in T1DM islets [516,517]. Activated M1 macrophages secrete exosomes that impair insulin secretion of β cells [39]. Persistent MEX-miRs signaling might be involved in the immune pathogenesis of T1DM.

### 7.2. Type 2 Diabetes Mellitus

Only few epidemiological studies have compared the risk of milk consumption versus fermented milk/products in relation to the risk of T2DM [518]. The Lifeline Cohort Study identified positive associations of full-fat dairy products, non-fermented dairy products, and milk with newly diagnosed T2DM [519]. Persistent uptake of bovine MEX miR-148a and miR-21 via continued consumption of pasteurized cow milk may increase the risk of T2DM [36,520,521,522,523,524]. In fact, significantly increased circulatory levels of miR-148a-3p, miR-148a-5p, and miR-21-5p have been detected in diabetic patients [176,520]. Serum levels of miR-148a-3p correlate with HbcA1 levels [176] and circulating miR-21-5p identifies the hyperglycemic state in high-risk subjects [525]. Overexpression of miR-21 in the INS-1 β cell line increased proliferation as well as apoptosis [526]. MiR-21 has been involved in mediating β cell dysfunction [527]. In fact, miR-21 is upregulated in human islets from glucose intolerant donors [528]. Moreover, miR-21 is upregulated approximately two-fold in islets from db/db diabetic mice [529], from ob/ob mice [530], and in neonatal rat pancreas, coinciding with the time point of maximal β cell proliferation rate [531]. MEX miR-21 may thus increase physiological β cell proliferation during the postnatal period but may de-differentiate adult β cells back to the proliferative phenotype.

Milk, a donor of branched-chain amino acids and anabolic miRs including miR-148a and miR-21, enhances mTORC1 activity [70,159]. Elevated mTORC1 activation is a striking pathogenic hallmark of islets in T2DM contributing to impaired β cell function [175,230]. MiR-148a attenuates the expression of AMPK components [231,232] and PTEN [160], and thus activates mTORC1 [159]. In contrast, sestrins activate AMPK [391] and, via AMPK and GATOR interaction, also suppress mTORC1 [392]. It has recently been shown in monocytes of T2DM patients that sestrin 2 plays a major role in regulating the AMPK-mTORC1-pathway [532]. Circulatory levels of sestrin 2 are significantly decreased in diabetic patients [533,534]. *SESN2* is a target of miR-148a [535]. Thus, MEX miR-148a signaling via uptake of pasteurized milk may promote the proliferative neonatal β cell phenotype with increased mTORC1 but compromised AMPK activity, pointing to a potential mechanism contributing to β cell de-differentiation [36,230].

Persistent overactivation of mTORC1 induces ER stress, which in the longer run promotes β cell apoptosis [536]. Hyperactivated mTORC1 impaired β cell autophagy, resulting in β cell failure [537,538]. Thus, the administration of bovine MEX miR-148a and miR-21 to T2DM patients via consumption of pasteurized milk may cause β cell de-differentiation as well as mTORC1-driven ER stress, promoting β cell failure and an early death.

### 7.3. Metformin

The most common antidiabetic drug—metformin—improves β cell differentiation and counteracts milk miR signaling. MEX miR-148a-mediated silencing of *DNMT1* with reduced H19 promoter methylation may enhance H19 expression [108]. In contrast, metformin via methylating the H19 promoter region reduces H19 expression [111,112,113,114,115]. Metformin-mediated H19 methylation may impair the access of c-MYC to the H19 site of transcription regulation, which may also affect access of E2F1. This mode of action may be central for explaining the antidiabetic effects of metformin [186,187]. Exposing the action of metformin cancer cells showed that metformin leads to hypermethylation of tumor-promoting pathway genes and the concomitant inhibition of cell proliferation. Remarkably, metformin upregulated let-7 through AMPK activation, leading to degradation of H19 [539].

Metformin is an activator of AMPK [188] and thus acts as an inhibitor of mTORC1 [189]. As shown in a breast cancer cell line, metformin reduced the expression of miR-21 and enhanced the expression of critical upstream activators of the AMPK including sestrin 1, leading to AMPK activation and inhibition of mTORC1 signaling [540]. Metformin also reduced circulatory miR-21 expression in patients with diabetic nephropathy [541]. Notably, metformin conferred protection against high glucose-induced apoptosis of pancreatic β cells by alleviating oxidative and ER stress-induced CD36 expression [542].

Taken together, metformin prevents β cell de-differentiation to the proliferative neonatal phenotype with high mTORC1 signaling by activating AMPK, suppressing H19 and miR-21 signaling, thus displaying the opposite of lactation-mediated MEX miR signaling. Moreover, metformin upregulates aquaporin 7 (*AQP7*) in pancreatic islets (INS-1 cells) damaged by hyperglycemic conditions through the suppression of p38 and JNK mitogen activated protein kinase signaling to promote glycerol influx into cells and subsequent promotion of insulin secretion [351,352]. Remarkably, *AQP7* is a predicted target of miR-148a [543]. MEX miR-148a signaling may attenuate AQP7-mediated insulin secretion in analogy to miR-148a-mediated inhibition of *ABCA1*.

## 8. Discussion

The previous knowledge obtained from animal studies underlines the importance of the immediate postnatal period of life for metabolic programming [544,545]. Notably, the birth-to-weaning period plays a key role for pancreatic β cell expansion and functional maturation essential for the appropriate adaptation of plasma insulin levels to metabolic needs in adult life. These events are driven by epigenetic programs triggered by growth factors, hormones, and nutrients [7,11]. Our review highlights the potential impact of milk-derived exosomal (MEX) miRs in postnatal epigenetic programming of β cells. Indirect evidence gathered in this review proposes a direct impact of MEX-mediated postnatal programming of β cells with mass expansion. The consecutive switch to β cells to functional maturity may result from the weaning-related loss of MEX miR signaling [36]. Our hypothesis is supported by recent observations of bidirectional exosome traffic of β cells communicating with various distant cells and tissues of the body [37,38,39,40,41,42,43,44,45,46,47]. Exosomal miRs emerged as pivotal regulators in the pathogenesis of diabetes [1,41,51,52,53,505,546]. MiRs confer robustness to biological processes determining cell fate decisions and maintaining cell differentiation [547]. In fact, miRs are regarded as key regulators of pancreatic β cells physiology, function, and failure [7,548,549,550,551]. Importantly, increasing evidence indicates that miRs contribute to the maintenance of β cell identity and are involved in β cell de-differentiation [552,553,554,555]. Key components of β cell failure in T2DM involves: (1) loss of cell identity, specifically a reduction of transcription factors associated with mature cell function (NEUROD1, FOXO1, MAFA, PDX-1, NKX2.2, NKX6.1), as well as (2) de-differentiation, defined by regression to a progenitor or stem cell-like state [555,556,557]. According to Moin and Butler [552], the endocrine identity of diabetic β cells recapitulates the less mature/less-differentiated fetal/neonatal cell type, which very well matches with the period of lactation.

Of interest, β cell de-differentiation and loss of β cell identity has been associated with increased islet amyloid deposits [558]. Recent evidence indicates that β cell amyloid formation in mice is promoted by α-synuclein [559]. Upregulation of α-synuclein expression is promoted by miR-148a in a DNMT1-dependent manner [524,560]. Furthermore, miR-21, which is significantly upregulated in diabetic β cells [525,526,527,528,529,530,531], targets lysosome-associated membrane protein type 2A (*LAMP2A*) and thus reduces chaperone-mediated autophagy [561,562], thus linking α-synuclein toxicity to the pathogenesis of T2DM and Parkinson’s disease [524,563].

There are good reasons to assume that MEX and their lactation-derived miRs communicate with the neonatal β cells to support mTORC1-dependent β cell proliferation, growth, and mass expansion favoring glycolysis, which offers sufficient precursor molecules for biosynthetic pathways comparable to the Warburg effect seen in proliferating embryonic and cancer cells. Furthermore, MEX miRs may delay β cell maturation and differentiation during the period of breastfeeding, a time period of life with a relatively constant oral intake of lactose. The translational evidence presented in this review allows for the prediction that the most abundant exosomal miR of human breast milk, miR-148a, drives mTORC1/c-MYC activation, but suppresses p53, AMPK, TGF-β-mediated β cell differentiation, ERRγ/PGC-1α-driven mitochondrial oxidative phosphorylation, glucose sensing, insulin vesicular traffic, and GSIS. Lactation-mediated MEX miR-148a signaling may thus execute transcriptional and epigenetic events supporting sufficient β cell mass acquisition during the breastfeeding period for appropriate β cell function and glucose responsiveness in adult life. We suggest that postnatal β cell maturation is not only associated with islet-specific intrinsic miR changes induced by nutrient shifts at weaning [12], but by the loss of MEX miR signaling at weaning.

Physiological MEX miR-148a signaling is vulnerable and can be negatively affected by maternal obesity, GDM, caesarean delivery, short duration of breastfeeding, administration of HoP-processed human donor milk, and especially by artificial infant formula feeding. It is of critical concern that formula misses all MEX miRs of human milk.

Obesity is a pandemic [564], and, according to the WHO, 39% of adults aged 18 years and over were overweight in 2016, and 13% were obese [565]. Maternal overweight and obesity apparently deteriorates the quality of maternal milk miRs and its capacity for adequate MEX miR-148a/miR-30b signaling [420]. Furthermore, adipocyte-derived exosomes of adipocytes interact with human β cells [42]. Notably, ATM-derived exosomes from obese mice suppressed insulin secretion and enhanced β cell proliferation [46], thus promoting a neonatal β cell phenotype. Obese individuals may thus be more vulnerable to MEX miR signaling via consumption of pasteurized milk.

Multiparity has been related to increase the risk of T2DM [566]. Lactogenic hormones of pregnancy increase β cell numbers as an adaptive response to enhanced insulin requirements [567]. Ageing is also associated with an increased risk of T2DM [568,569]. In elderly women, TGF-β receptor 1 expression correlated negatively with miR-21 expression in PBMCs [570].

According to WHO, caesarean section rates continue to rise globally [571]. In Latin America and the Caribbean, rates are as high as 43% of all births. In five countries (Dominican Republic, Brazil, Cyprus, Egypt, and Turkey), caesarean sections now outnumber vaginal deliveries [571]. Caesarean section is a further critical risk factor reducing MEX miR-148a and miR-125b levels in human milk [153]. Notably, a population-based registry study of 2,699,479 births in Denmark from 1973 to 2016 showed an increased risk of diabetes 40 years after caesarean section compared to vaginal delivery [572]. Thus, an unjustified caesarean section may increase the risk of T2DM later in life, potentially due to adverse effects of MEX miR signaling, impairing postnatal β cell proliferation.

A century ago, the mode of infant feeding has changed from “breast to bottle” with the erroneous idea that human milk is “just food” [573]. In 1929, Marriott and Schoenthal [574] recommended unsweetened evaporated milk for general use as milk for infants. Due to the vast gap of knowledge of the molecular biology of milk, this historical misinterpretation completely ignored milk’s capacity for epigenetic programming of the infant [13,14,19,157,463,575,576]. The exaggerated upregulation of FTO (target of miR-30b) and CPT1A (target of miR-148a) in blood monocytes of formula fed infants compared to breastfed controls [474], clearly shows that formula is unable to meet the appropriate transcriptional and epigenetic axes naturally provided by breastfeeding, which is the gold standard controlled by the human lactation genome.

A century ago, large-scale pasteurization of cow milk, a relative gentle thermal processing compared to boiling or ultraheat treatment, was introduced to restrict the growth of harmful pathogenic bacteria in raw milk [577,578]. The detection of EVs and exosomes in human and bovine milk is a very recent insight. Now, we have learned that bovine milk EVs and exosomes and their miR cargo, especially miR-148a and miR-21, survive pasteurization [475,476]. Bovine MEX derived from pasteurized commercial milk enter the circulation and reach distant organs, as shown in rodent models [30]. The major properties of exosomes seem to have been conserved over eons, suggesting that they may have ancient evolutionary origins [236,579]. In fact, the top 10 miRs of milk of various mammalian species are highly conserved and exhibit identical nucleotide sequences between *Homo sapiens* and *Bos taurus* [15,21]. Recent evidence underlines that oral administration of exosomal bovine miR-148a is bioavailable and reaches target tissues (liver, brain) of C57BL6/ mice [28]. Continued exposure of humans to pasteurized cow milk and their MEX miRs may exert gene-regulatory effects [272,354,580] that may also affect the pancreatic islet of the milk consumer [36,519,520,521,522,523,524]. The adult human consumer of pasteurized cow milk is thus continuously exposed to bovine MEX including miR-148a and miR-21, which may increase mTORC1 signaling [159], promoting mTORC1-driven β cell de-differentiation [36,230].

Whereas exosome miR may be valuable tools for the early monitoring of T2DM [1], the use of bovine MEX for the supposed “improvement of nutrition” and many therapeutic options [581,582,583,584,585], may be another critical experiment on humans, as the operational period of MEX miRs is physiologically restricted to the postnatal period. The artificial use of MEX beyond the period of breastfeeding may turn the highly valuable MEX for the infant [14,15,16,17,18,19,586,587,588,589] into diabetogenic pathogens in adult life [36,520,521,522,523,524].

## 9. Conclusions

We conclude that all efforts should be undertaken to promote breastfeeding, allowing for adequate undisturbed MEX miR signaling during postnatal life to support adequate postnatal β cell development. In contrast to their postnatal importance, during adult life, bovine MEX should not reach the human milk consumer to avoid MEX miR-mediated β cell de-differentiation [36,523].

We are very concerned about the recent euphoric promotion of bovine MEX for purposes of human nutrition as well as systemic administration for medical treatments [353,584,585,586,587,588,589]. In contrast to García-Martínez et al. [582], who praise the generally “beneficial effects” of bovine MEX in metabolic interorgan cross-talk, we only see a meaningful potential use of MEX for infants of mothers who cannot provide adequate breastfeeding and depend on artificial formula administration. We must ask: What happens to the pancreatic β cells when bovine MEX are systemically administered over prolonged treatment periods with the intention to improve other chronic degenerative diseases such as osteoarthritis or osteoporosis [590,591,592], chronic inflammatory intestinal diseases [593,594,595,596], or chronic states of tissue fibrosis [597,598]?

Mammalian evolution provides MEX exclusively for the period of lactation and metabolic regulations that are operative during this restricted period at the beginning of mammalian life until weaning. Figure 9 illustrates the various constellations of MEX miR signaling during the postnatal period.

After the weaning period, the postnatal beneficial actions of MEX may turn into diabetogenic pathogens, as persistent MEX miR signaling may promote β de-differentiation back to the neonatal β cell phenotype. As recently suggested by Askenase [599], new experiments on molecular and quantitative miR functional effects in systems that include EVs, such as variation in EV type and surface constituents, delivery, dose, and time to hopefully create more appropriate and truly current canonical concepts of the consequent miR functional transfers by EVs including exosomes. This is of course of key importance to understand the physiological and pathological interactions of human MEX as well as bovine MEX with the human β cell.

Our presented model of MEX signaling in neonatal and adult life predicts the beneficial diabetes-preventive effects of MEX for the infant but “chronic MEX toxicity” through persistent delivery of bovine MEX that may converge with further exosome-mediated adverse effects of glucotoxicity, lipotoxicity, and chronic inflammation.

## 10. Limitations

We would like to note that a weak point of our presented hypothesis on the potential role of MEX miR-related effects on postnatal β cell proliferation and β cell de-differentiation in adult life is the fact that evidence is not exclusively derived from experimental data obtained from β cells. Furthermore, we would like to mention that MEX not only deliver miRs, and that miRs are abundant and diverse in milk. They are detected in fractions unrelated to MEX, including exfoliated cells, lipid, and protein fractions, which may as well contribute to β cell differentiation or de-differentiation. In addition, MEX are not always loaded with miRs, and a high amount of empty particles have been described. Exosomes can also harbor miRs, for instance, stuck onto Toll-like receptors at their surface. MEX may carry biological information unrelated to non-coding RNAs modulating host physiology. We selected the most abundant signature MEX miRs of human and bovine milk such as miR-148a and miR-21 to promote our hypothesis. Less highly expressed MEX miRs may also strongly influence β cell physiology and pathology, which at present is out of range of scientific imagination and may be settled by future research supported by artificial intelligence.

## 11. Materials and Methods

We conducted our bibliographic research exclusively via PubMed using various keywords such as “beta-cell”, “β cell”, “postnatal β cell development”, “β cell development”, “postnatal β cell mass”, “β cell maturation”, “nutritional switch during weaning”, “diabetes predisposition”, “fetal environment and diabetes risk”, “maternal obesity”, “gestational diabetes mellitus”, “maternal undernutrition”, “transgenerational inheritance of diabetes”, “exosomes”, “milk exosomes”, “extracellular vesicles”, “microRNA”, “miRNA”, and “miR”. We also analyzed papers based on the bibliographic references cited by the studies found on PubMed during our search. Whenever possible, we selected the most recent and comprehensive reviews on the topic in question. All selected articles were written in English.

## Figures and Tables

**Figure 1 ijms-23-11503-f001:**
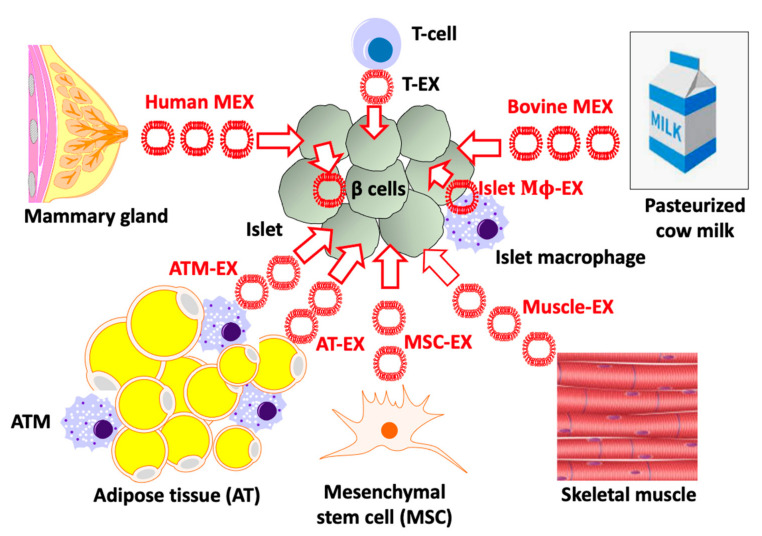
Illustration of exosome crosstalk to pancreatic β cells derived from various organs and exosome-emitting cells. Exosomes of human milk (MEX) during the postnatal breastfeeding period as well as bovine MEX derived from pasteurized cow milk may reach the islet cells of the milk recipient modifying β cell gene regulation. Abbreviations: AT = adipose tissue; ATM = adipose tissue macrophage; Islet Mϕ-EX = Islet macrophage-derived exosome; MEX = milk-derived exosome; MSC-EX = MSC-derived exosome; Muscle-EX = Skeletal muscle cell-derived exosome; T-EX = T cell-derived exosome.

**Figure 2 ijms-23-11503-f002:**
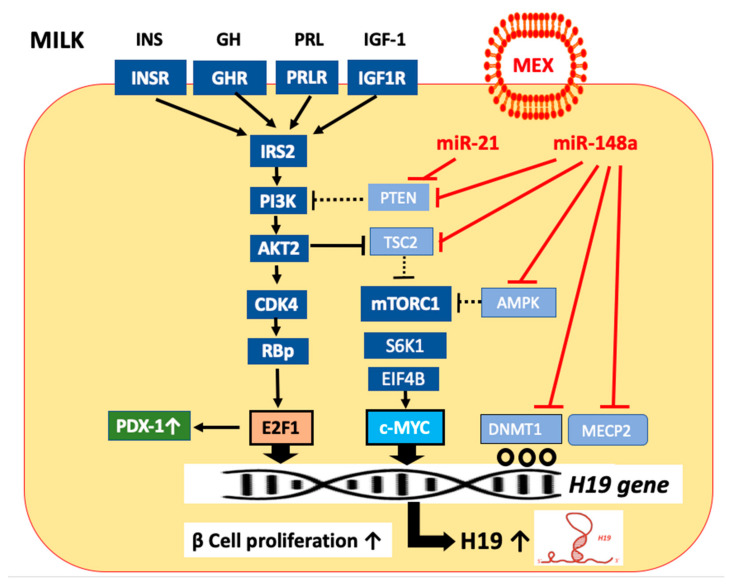
Milk may induce AKT2/mTORC1-mediated β cell proliferation. The growth-promoting hormones of milk, i.e., insulin (INS), growth hormone (GH), prolactin (PRL), and insulin-like growth factor 1 (IGF-1) synergistically activate the kinases AKT and mechanistic target of rapamycin complex 1 (mTORC1). Key exosomal milk (MEX) miRs of human and bovine milk (miR-148a and miR-21) may further augment AKT-mTORC1 signaling by targeting PTEN, TSC2, and AMPK subunits, resulting in increased expression of the proliferative transcription factors E2F1 and c-MYC, respectively. MEX miR-148a-mediated suppression of DNMT1 and MECP2 may decrease H19 promoter methylation, enhancing the transcription of H19 (indicated as open circles).

**Figure 3 ijms-23-11503-f003:**
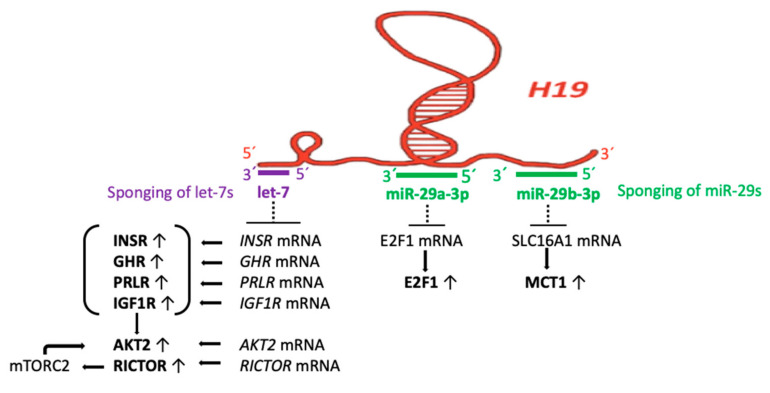
Long non-coding RNA H19 augments β cell proliferation. H19 functions as a sponge of let-7 miRs and miR-29 family members, resulting in the enhanced expression of growth hormone receptors, AKT2, and the mTORC2 component RICTOR. Growth hormone receptors and mTORC2 activate AKT2 and its downstream target mTORC1. H19-mediated sequestration of miR-29a-3p enhances the expression of E2F1, a known transcription factor promoting H19 expression. H19-mediated-sponging of miR-29b-3p enhances the expression of monocarboxylic acid transporter 1 (MCT1 encoded on *SLC16A1*), a disallowed gene of mature β cells. Abbreviations: INSR = insulin receptor; GHR = growth hormone receptor; PRLR = prolactin receptor; IGF1R = insulin-like growth factor 1 receptor; AKT2 = AKT serine/threonine kinase 2; RICTOR = rapamycin-insensitive companion of mTOR; E2F1 = E2F transcription factor 1.

**Figure 4 ijms-23-11503-f004:**
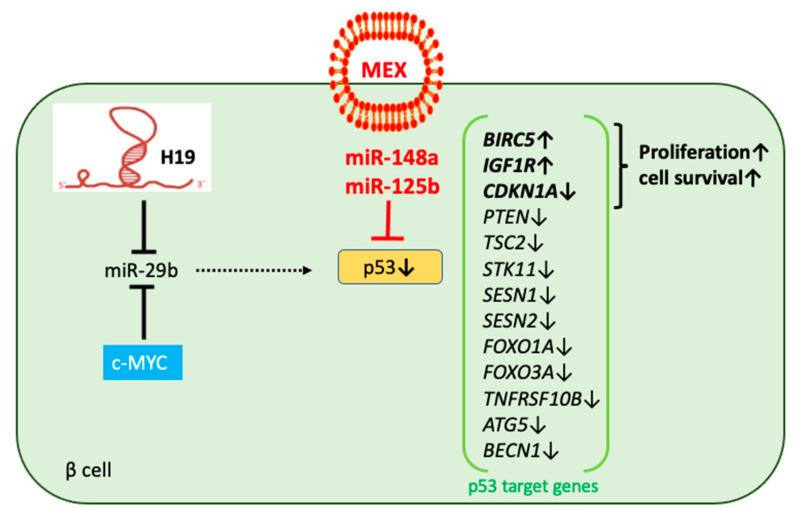
The predicted impact of milk miR signaling on β cell p53-dependent gene expression. Milk exosome (MEX)-derived miR-148a and miR-125b are abundant miRs of human and bovine milk that target p53. Reduced p53 signaling may modify genes involved in cell proliferation (*CDKN1A*, *IGF1R*) and survival (*BIRC5*). Enhanced expression of long non-coding RNA H19 and transcription factor c-MYC reduces the expression of miR-29b, a positive regulator of p53 expression.

**Figure 5 ijms-23-11503-f005:**
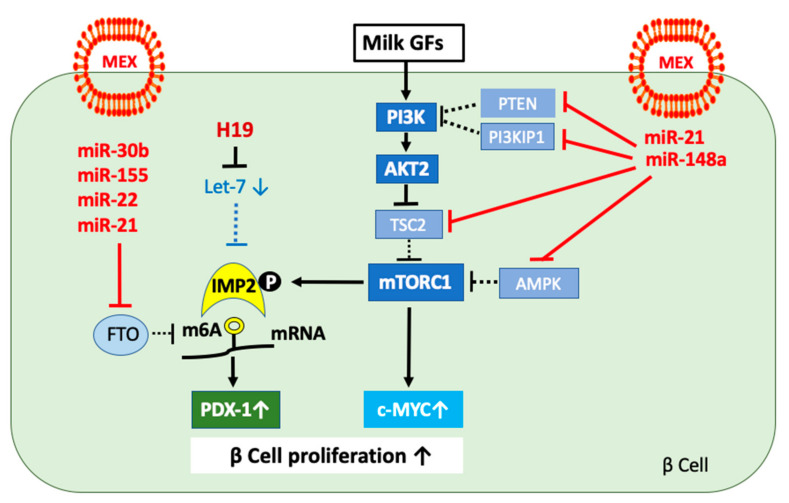
Potential impact of milk exosome (MEX) miR-mediated N6-methyladenosine (m6A) RNA modification of pancreas/duodenum homeobox protein 1 (PDX-1). MEX miR-induced suppression of fat mass- and obesity-associated gene (*FTO*) may stabilize m6A on *PDX1* mRNA, enhancing PDX-1 expression and promoting β cell proliferation. MEX miR-stimulated activation of mTORC1-mediated phosphorylation of insulin-like growth factor 2 mRNA-binding protein 2 (IMP2) as well as H19-mediated sponging of let-7 may further augment PDX-1 expression, supporting β cell proliferation.

**Figure 6 ijms-23-11503-f006:**
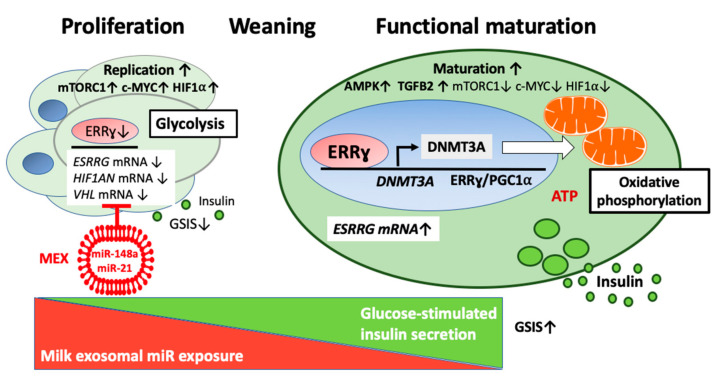
Predicted model of milk exosome (MEX) miR-mediated suppression of estrogen-related receptor γ (ERRγ). MiR-148a targets *ESRRG* mRNA and thus impairs ERRγ expression, a negative regulator of glycolysis. MiR-148a and miR-21 promote glycolysis via targeting hypoxia-inducible factor 1α inhibitor (*HIF1AN*) and von Hippel-Lindau tumor suppressor (*VHL*), respectively. Both are key negative regulators of hypoxia-inducible factor 1α (HIF-1α), the key transcription factor of glycolysis, which is required for cell proliferation. Functional maturation of β cells after weaning is associated with fading MEX miR signaling, enhancing the expression of ERRγ, the key transcription factor of mitochondrial oxidative phosphorylation. ERRγ may also promote the expression of DNA methyltransferase 3A (*DNMT3A*), which, via promoter methylation, may suppress “disallowed genes” involved in glycolysis. ERRγ/DNMT3A signaling may stimulate insulin secretion coupled to glucose levels.

**Figure 7 ijms-23-11503-f007:**
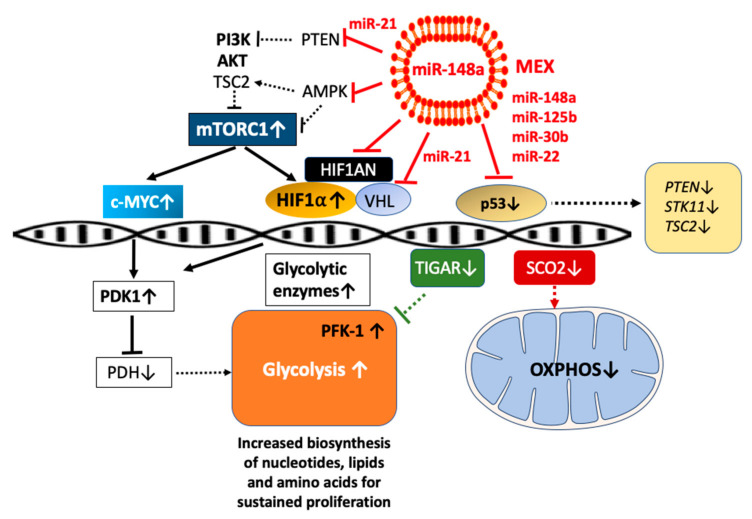
Predicted model of MEX miR-mediated regulation on glycolysis and mitochondrial oxidative phosphorylation (OXPHOS). Upregulated mTORC1 signaling activates the expression of c-MYC and hypoxia-inducible factor 1α (HIF-1α), which both upregulate the expression of pyruvate dehydrogenase kinase 1 (PDK1), resulting in an inhibition of pyruvate dehydrogenase (PDH), thereby activating glycolysis. Glycolysis provides key substrates of precursor molecules for biosynthetic pathways required for cell proliferation. MiR-148a and miR-21 promote glycolysis via targeting hypoxia-inducible factor 1α inhibitor (*HIF1AN*) and von Hippel-Lindau tumor suppressor (*VHL*), respectively. Both are pivotal negative regulators of HIF-1α, the key transcription factor of glycolysis. MEX miR-mediated suppression of p53 attenuates the expression of TP53-induced glycolysis and apoptosis regulator (*TIGAR*), thereby enhancing the activity of 6-phoshofructo-1 kinase (PFK-1), which promotes glycolysis. In addition, MEX miR-mediated suppression of p53 also reduces the expression of SCO cytochrome c oxidase assembly protein 2 (*SCO2*), which operates at the inner membrane of the mitochondria, facilitating the assembly of cytochrome c oxidase complex in the electron transport chain. Thus, MEX miR signaling promotes glycolysis but attenuates mitochondrial activities during the period of lactation-induced β cell proliferation, a meaningful mechanism for the postnatal period, which fades with weaning.

**Figure 8 ijms-23-11503-f008:**
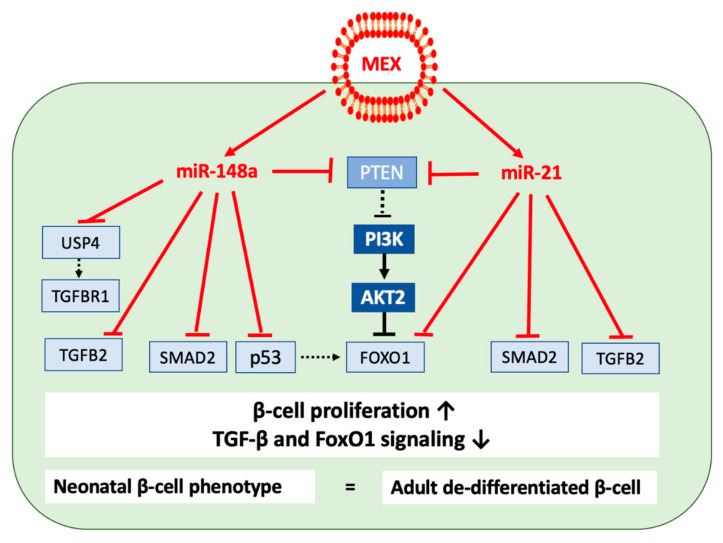
Illustration of the involvement of MEX miRs on β cell transforming growth factor β (TGF-β) and forkhead box O1 (FOXO1) signaling. MEX miR-148a and miR-21 attenuate TGF-β signaling, which plays a key role in β cell differentiation. MiR-148a targets ubiquitin-specific protease 4 (*USP4*), which de-ubiquitinates TGF-β type 1 receptor (TGFBR1). MiR-148a directly targets transforming growth factor β2 (*TGFB2*) and *SMAD2*, and thus attenuates TGF-β signaling at various checkpoints. MiR-21 targets *FOXO1A*, *SMAD2,* and *TGFB2*, which may also impair FoxO1/SMAD-regulated gene expression. Weaning-associated fading of MEX miRs may thus enhance TGF-β—and FoxO1 signaling, promoting β cell differentiation and cell cycle arrest. Continued MEX exposure may thus contribute to β cell de-differentiation.

**Figure 9 ijms-23-11503-f009:**
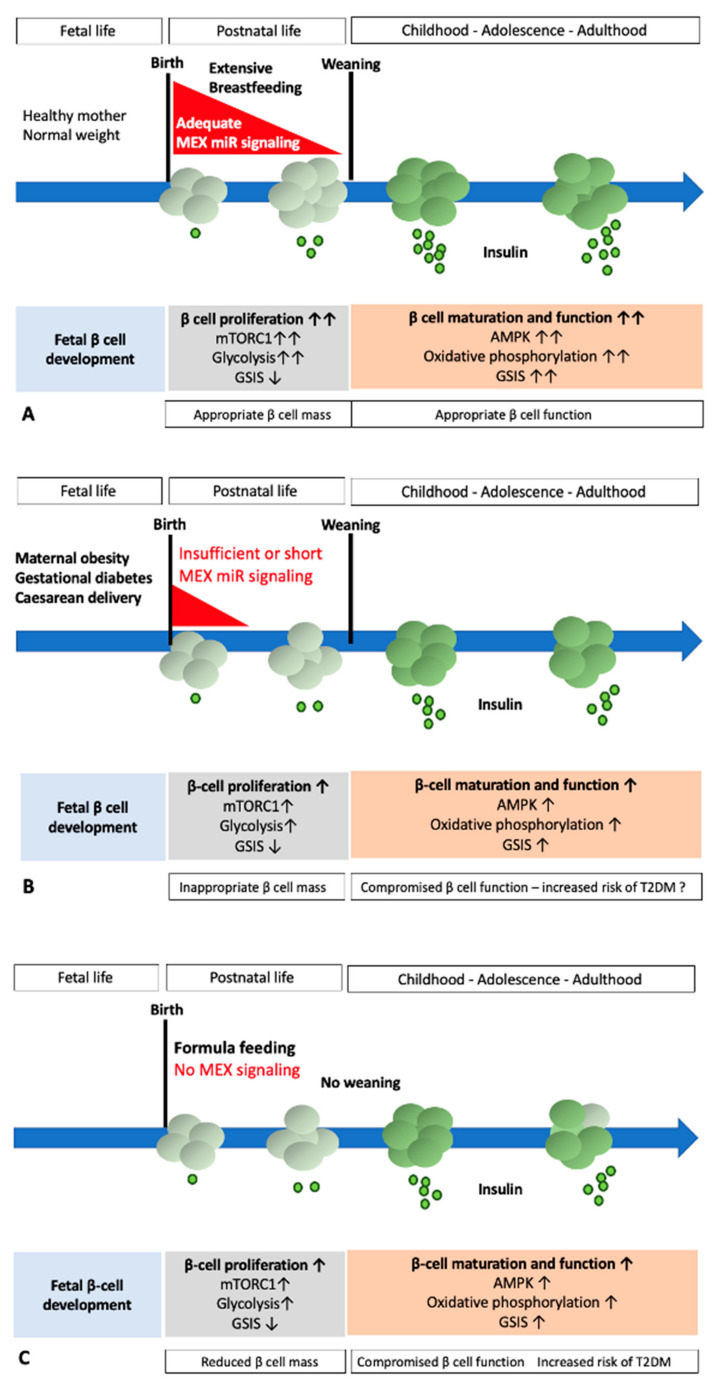
Selected constellation of milk exosome (MEX) miR signaling during the postnatal period. (**A**) Physiological breastfeeding promoting adequate β cell mass expansion. No further bovine MEX exposure after the lactation period. (**B**) Insufficient or disturbed postnatal MEX signaling due to maternal obesity, gestational diabetes, or caesarean delivery, compromising β cell mass expansion. No further bovine MEX exposure during adult life. (**C**) Artificial postnatal formula feeding without MEX signaling and no further MEX availability during adult life. One arrow up = upregualtion; two arrows up = extensive upregulation; one arrow down = downregulation.

**Table 1 ijms-23-11503-t001:** Potential gene-regulatory effects of MEX miR-148a on postnatal β cells.

miR-148a Target Genes	Predicted Gene-Regulatory Effects of MEX miR-148a	References
*TP53*	Increased expression of survivin (BIRC5) and suppression of p21 (CDKN1A) stimulates β cell proliferation; decreased expression of TSC2 activates mTORC1; reduced sestrins (SESN1, SESN2) and LKB (STK11) expression attenuates AMPK activity enhancing mTORC1 activation; decreased expression of miR-29 with diminished suppression of MCT1; decreased expression of FoxO1, which controls β cell differentiation/identity	[132]
*DNMT1*	Reduced expression of DNMT1 results in lower DNA methylation of gene promoter regions, increasing general transcription	[15,160]
*PTEN*	Reduced expression of PTEN activates PI3K and thus AKT-mTORC1	[160]
*MECP2*	Reduced expression of MECP2 attenuates DNMT1-mediated DNA methylation, enhancing gene expression	[183]
*PRKAA1*	Reduced AMPK activity increases mTORC1 signaling	[231]
*PRKAG2*	Reduced AMPK activity increases mTORC1 signaling	[232]
*TSC2*	Reduced expression of TSC2 enhances RHEB-mediated mTORC1 activation	[282]
*ESRRG*	Reduced expression of ERRγ inhibits the mitochondrial function and oxidative phosphorylation required for GSIS	[258]
*PPARGC1A*	Suppression of mitochondrial biogenesis, oxidative phosphorylation, and GSIS	[241,242]
*CPT1A*	Reduced transfer of free fatty acids to mitochondrial oxidative phosphorylation	[245]
*HIF1AN*	Reduced suppression HIF-1α enhances glycolysis for the promotion of β cell growth	[296]
*MAFB*	Reduced expression of MAFB results in reduced expression of GLUT1 and GLUT2	[313]
*NEUROD1*	Attenuation of β cell differentiation and insulin gene expression	[347]
*SLC2A1*	Reduced expression of GLUT1 decreases glucose sensing	[317]
*SMAD2*	Reduced SMAD2-mediated TGF-β signaling	[334]
*MAX*	Reduced c-MYC/MAX-mediated promoter activation of pyruvate kinase (PK) expression attenuates PK-controlled GSIS	[180]
*ABCA1*	Impaired insulin granule function and insulin secretion	[324]
*USP4*	Increased proteasomal degradation of TGFBR1 attenuates TGF-β signaling	[335,336,337]
*AQP7*	Reduced AQP7-mediated insulin secretion	[351,352]

**Table 2 ijms-23-11503-t002:** Perinatal factors modifying MEX miR signaling.

Impact Factors	Effects on MEX miR Concentrations	References
Maternal obesity	miR-148a ↓ miR-30b ↓	[417]
Gestational diabetes mellitus	miR-148a ↓ miR-30b ↓	[429]
Maternal stress during pregnancy	miR-155 ↑ miR-96 ↑	[402]
Caesarean delivery	miR-148a ↓ miR-125b ↓	[153]
Oxytocin treatment	miR-148a ↑ miR-30 ↑ miR-320 ↓	[438]
Holder pasteurized human donor milk	miR-148a ↓	[457]
Preterm birth	miR-148a ↑ miR-22 ↑	[182,370]
Infant formula	miR-148a ↓↓ (below detection limit)	[153,158,458]
Soy protein-based formula	Impaired intestinal MEX miR uptake	[459]
Raw cow milk	miR-148a ↑↑ miR-21 ↑↑	[158,475]
Pasteurized cow milk	miR-148a ↑ miR-21 ↑ miR-30d ↑	[475,476]
Ultraheat-treated cow milk	miR-148a and others miRs ↓↓	[475,476]

One arrow up = upregualtion; two arrows up = extensive upregulation; one arrow down = downregualtion; two arrows down = extensive downregulation.

## Data Availability

Not applicable.

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
