# Peer review of "Milk Exosomal microRNAs: Postnatal Promoters of β Cell Proliferation but Potential Inducers of β Cell De-Differentiation in Adult Life"

_ijms, 2022, doi:10.3390/ijms231911503_

Round 1
Reviewer 1 Report
This a highly important review that attempts to highlight the potential role of milk as carrier of exosomal microRNAs (MEX miRs). The authors do a good job of using existing literature to hypothesize a potential role of MEX miRs in post-natal beta cell proliferation and function.
The central theme of the review is that MEX miRs promote postnatal beta cell proliferation at the expense of beta cell maturation. Based on this logic, the authors present the argument for the benefits of MEX miRs for increasing in beta cell mass during the period of breastfeeding. Conversely they argue how milk consumption past weaning could contribute to diabetes through MEX miR mediated beta cell dedifferentiation.
This subject is of high public health relevance, and the authors are commended for the diligent effort to put together such a compelling review. However, in the current state, the manuscript is too dense. Throughout the manuscript, some of the presented information appear superflous. This makes it extremely difficult to read and appreciate the central message.
The authors need to reduce the information load, and reorganize the writing to be more concise.
This reviewer has chosen to use 2 examples to illustrate the difficulty of assimilating the information presented.
1) In section 3.5, the underlying message appears to be that weaning correlates with increased expression of p53 and miR-29, and both factors are engaged in a positive feed-back loop. But there is no clear link to how the absence of milk derived MEX miRs contribute to the expression of the above 2 factors. Also the relevance of p53 and mir-29 expression post weaning is not clear. This lack of clarity might be due to excessive information. For instance the information from lines 289 to 296 appear to detract from the main message.
Perhaps reorganizing the sequence of presentation might improve clarity. For instance if the authors opened section 3.5 by highlighting the inverse relationship between beta cell proliferation and maturation (as in lines 422 to 428). Then briefly introducing pro-proliferative factors such as AKT/c-myc versus p53 as anti-proliferative factor (and therefore enhancing beta cell proliferation) would suffice. Such an introduction would make it easier to appreciate the physiological relevance of specific MEX miRs that influence p53 or AKT (leading to c-myc).
2) Another example is 3.4 (LIN28A and LIN28B). Substantial information is presented about LIN28A and LIN28B that does not appear directly contribute to the key message. This section clearly suggests that during breastfeeding, milk stimulates beta cell proliferation through H19 mediated suppression of let-7. But the link between let-7 and LIN28A and LIN28B is not clear. One sentence (line 263-264) indicates that LIN28A and LIN28B are targets of let-7, but it lack literature reference. Without this clear link, the other information about LIN28A and LIN28B, at the beginning of the section, appear irrelevant and distracting from the main message.
It is the hope of this reviewer that the authors would make an effort to increase the clarity of this important review article for the benefit of a wider readership.
Author Response
Response to Reviewer 1
We thank R1 for the appreciation of our invited manuscript, which intends to present a comprehensive review of the milk exosome (MEX)-derived microRNAs (miRs) during the postnatal period of natural breastfeeding as well as the potential impact of cow milk-derived MEX miR signaling on adult beta-cells potentially promoting beta-cell de-differentiation back to the postnatal proliferating phenotype.
R1 is concerned by the high information load, which is indeed excessive, but covers the whole spectrum of available MEX miR data related to beta-cell physiology. We modified subtitles and reduced redundant text throughout the manuscript to further improve the clarity of the manuscript, which apparently was not a matter of concern for R2.
Section 3.5 (p53, miR29)
We improved this section to provide clearer information according to p53 and miR-29 and their interactions in relation to beta-cell homeostasis. All changes have been labeled in yellow.
Section 3.4 (LIN28)
Furthermore, we improved this section on the role of LIN28 in let-7 suppression, which synergizes with H19-mediated let-7 suppression, which is of critical importance for growth factor signaling enhancing beta-cell proliferation.
We thank R1 for all valuable comments and hope that our changes improved the revised version of the manuscript.

Reviewer 2 Report
The manuscript « Milk exosomal microRNAs : postnatal promoters of beta-cell proliferation but potential inducers of beta-cell de-differentiation in adult life » is well in line with the Special Issue "Characterization and Biological Function of Milk-Derived miRNAs", under the Section « Bioactives and Nutraceuticals ».
The review can be published in Int J Mol Sciences after some corrections if the journal is accepting speculative papers. It would be better to publish it under « Opinion paper » or the like. But if the special issue is not offering this possibility, the review section will be OK.
The advantage of the paper of Melnik and Schmitz is that it provides a clear hypothesis, quoting 600 papers related to miRNAs and breast milk, from humans to many mammals. The illustrations are of quality and related to the current state of the art regarding biochemical pathways.
The authors are discussing an audacious hypothesis. The first weak point is clearly that authors are grouping papers unrelated to the physiology of beta-cell of the pancreas in order to fit into their hypothesis, frequently using mere correlation as proof. This is only a warning statement.
The second weak point is that the purpose of milk exosomes seems limited to delivering miRNAs cargo to immune and epithelial cells of the intestine.
Consequently, I would ask the authors to introduce some level of doubt on their very clear point of view along the following remarks 1 to 4.
(1) the miRNAs are abundant and diverse in milk. They are detected in fractions unrelated to exosomes (exfoliated cells, lipid and protein fractions) which can contribute to beta-cell differentiation or de-differentiation,
(2) exosomes are not always loaded with miRNAs, a high amount of empty particles have been described, and they can also harbor miRNA, for instance, stuck onto Toll-like receptors at their surface,
(3) exosomes are carrying biological information unrelated to non-coding RNAs which modulate the host physiology,
(4) authors are selecting only the highly expressed miRNAs, impeding research done on less abundant miRNAs which may reveal, in the future, to be more potent than highly expressed milk miRNAs.
I suggest that authors introduce a « limitation section » in their paper. I believe that it is highly probable that the milk exosomes are involved in mother and infant molecular communication, but the review should not disseminate that only miR-148 and miR-21 are actors in this communication.
(5) The panel's A to C of Figure 9 are sufficient to carry on the message of the paper. Panel D has to be removed because it implies nutritional advice (suppressing milk products) which is out of the scope of the review. In addition, this will help to enlarge Figure 9 which is far too small on my pdf copy.
Minor alterations :
(1) - line 649. « and reduced oxygen consumption phenocopying p53-deficient cells [285] ». The meaning of phenocopying, is is not clear. Thanks to clarify.
(2) I am listing below some sections that could be removed to shorten a long text. These are only proposals.
- Page 23. Section 6. 3. Maternal obesity. Please consider using the data elsewhere and suppress this section, which is not related to beta-cell mass.
- Page 24. Section 6.5 Maternal diet. Please consider using the data elsewhere and suppress this section that is unrelated to beta-cell mass.
- Page 24. Section 6.7 Changes of miR level during breastfeeding. Please consider using the data elsewhere and suppress this section, which is not related to beta-cell mass.
- Page 25. Section 6.9 Donor milk. Please consider using the data elsewhere and suppress this section that is unrelated to beta-cell mass.
Author Response
Response to Reviewer 2
General remarks:
We thank R2 for the appreciation of our manuscript, which intends to provide indirect and translational evidence for the potential role of milk exosomes (MEX) and their most abundant microRNAs (miRs) for postnatal and adult beta-cell homeostasis. We are aware that our view is still hypothetical and we agree to implement the section “Limitations” at the end of the manuscript (chapter 11) to discuss weaknesses and limitations of our review.
Figure 9
We agree that Figure 9 is too extensive and removed part D from the figure including changes of the corresponding figure legend.
Minor alterations:
We condensed the sections under subheading 6. These chapters exhibit the potential impact of maternal and iatrogenic factors on MEX miR composition, which are of importance for postnatal beta-cell proliferation and mass expansion. These sections may be of critical importance for clinicians and researchers interested in deviations of perinatal and postnatal epigenetic programming including the beta-cells physiology and pathology.
We appreciate all valuable remarks of R2 and hope that our revised version of the manuscript is now acceptable for R2.

Round 2
Reviewer 1 Report
The authors' have made satisfactory edits.